# Conserved conformational selection mechanism of Hsp70 chaperone-substrate interactions

**Ashok Sekhar[1,2,3]\*, Algirdas Velyvis[1,2,3], Guy Zoltsman[4], Rina Rosenzweig[1,2,3,4], Guillaume Bouvignies[5,6], Lewis E Kay[1,2,3,7]\***

[1]Department of Molecular Genetics, University of Toronto, Toronto, Canada; [2]Department of Chemistry, University of Toronto, Toronto, Canada; [3]Department of Biochemistry, University of Toronto, Toronto, Canada; [4]Department of Structural Biology, Weizmann Institute of Science, Rehovot, Israel; [5]Laboratoire des Biomolécules, Département de chimie, École normale supérieure, UPMC Univ. Paris 06, CNRS, PSL Research University, Paris, France; [6]Sorbonne Universités, UPMC Univ. Paris 06, École normale supérieure, CNRS, Laboratoire des Biomolécules, Paris, France; [7]Hospital for Sick Children, Program in Molecular Medicine, University Avenue, Toronto, Canada

**Abstract** Molecular recognition is integral to biological function and frequently involves preferred binding of a molecule to one of several exchanging ligand conformations in solution. In such a process the bound structure can be selected from the ensemble of interconverting ligands *a priori* (conformational selection, CS) or may form once the ligand is bound (induced fit, IF). Here we focus on the ubiquitous and conserved Hsp70 chaperone which oversees the integrity of the cellular proteome through its ATP-dependent interaction with client proteins. We directly quantify the flux along CS and IF pathways using solution NMR spectroscopy that exploits a methyl TROSY effect and selective isotope-labeling methodologies. Our measurements establish that both bacterial and human Hsp70 chaperones interact with clients by selecting the unfolded state from a pre-existing array of interconverting structures, suggesting a conserved mode of client recognition among Hsp70s and highlighting the importance of molecular dynamics in this recognition event.
DOI: https://doi.org/10.7554/eLife.32764.001

**\*For correspondence:**
ashoksekhar@yahoo.com (AS);
kay@pound.med.utoronto.ca
(LEK)

## Introduction

An elaborate network of chaperones is present in organisms across all domains of life to oversee the health of the cellular proteome (*Balchin et al., 2016*). The Hsp70 family of molecular chaperones occupies a central node in this network, steering proteins synthesized on the ribosome to their native conformations, as well as cooperating with the machinery of protein disaggregation, refolding and proteolysis to control the fate of improperly folded and aggregated cellular proteins (*Mayer and Bukau, 2005*; *Mayer, 2013*; *Mogk et al., 2015*; *Rosenzweig et al., 2013*). Hsp70 interacts with client substrates via an ATP-dependent chaperone cycle that is tightly regulated by Hsp40 co-chaperones and nucleotide exchange factors (NEFs) (*Balchin et al., 2016*; *Mayer and Bukau, 2005*; *Mayer, 2013*). Substrates are brought to the cycle by either Hsp40 or ATP-bound Hsp70 and typically undergo multiple rounds of binding and release before either folding to the native state or being transferred to downstream chaperone systems such as GroEL/ES and ClpB (bacteria) or Hsp90 and Hsp104 (eukaryotes).

**eLife digest** Proteins are the workhorses of a cell and are involved in almost all biological processes. Newly made proteins need to 'fold' into precise three-dimensional shapes in order to carry out their roles. However, proteins sometimes fold incorrectly or unfold. These protein forms are not able to work effectively and in some cases may even cause diseases.

Chaperone proteins help other proteins to fold correctly and are found in living organisms ranging in complexity from bacteria to humans. There are many different types of chaperones that play different roles inside cells. One, called Hsp70, binds to proteins that are incorrectly folded to help them to mature into their correct structures. However, it was not clear whether Hsp70 can also associate with the mature, correctly folded form of the proteins.

A technique called Nuclear Magnetic Resonance (NMR) spectroscopy can distinguish between mature, unfolded and chaperone-bound forms of the same protein. Sekhar et al. therefore used NMR to investigate which forms of a protein Hsp70 binds to. This revealed that both the bacterial and human versions of the Hsp70 chaperone interact only with unfolded proteins.

The results presented by Sekhar et al. also explain why Hsp70 does not disrupt the routine workings of the cell: because it does not bind to mature forms of proteins. These observations extend our understanding of how chaperones assist in folding proteins, and fit into a broader research theme exploring how proteins recognize one another. It will now be interesting to see whether the same mechanism holds for more complex forms of proteins, such as aggregates, or larger protein structures with regions of both folded and unfolded elements.
DOI: https://doi.org/10.7554/eLife.32764.002

Hsp70 is a 70 kDa protein that consists of a 45 kDa N-terminal ATPase domain (NBD) and a 25 kDa C-terminal substrate binding domain (SBD) (*Figure 1A*) (*Mayer and Bukau, 2005*). Much of our understanding of client recognition by Hsp70 derives from studies of peptide-based substrates. Crystal (*Zhu et al., 1996*; *Zahn et al., 2013*) and NMR structures (*Stevens et al., 2003*) of peptide-bound SBDs establish that the substrate binds in an extended conformation, with a 4–5 amino acid core of the substrate forming backbone hydrogen bonds as well as hydrophobic contacts with chaperone residues Val 436, Ile 401 and Ile 438 that line the SBD binding pocket (*Figure 1B*). The hydrophobic nature of the central binding pocket governs the preference of Hsp70 for substrate sequences enriched in large aliphatic hydrophobic sidechains such as Ile, Leu and Val (*Rüdiger et al., 1997*). Studies on full length protein substrates have revealed that these are globally unfolded in the Hsp70-bound state (*Palleros et al., 1994*; *Kurt et al., 2006*; *Chen et al., 2006*; *Kurt and Cavagnero, 2008*; *Sharma et al., 2010*; *Kellner et al., 2014*; *Lee et al., 2015*; *Sekhar et al., 2015*) and lack long-range interactions (*Sekhar et al., 2016*), though they can form local residual native (*Sekhar et al., 2015*) and non-native secondary structure (*Kurt and Cavagnero, 2008*) in regions far from the Hsp70 binding site. The ability of Hsp70 to recognize commonly occurring sequences in typical proteins with a frequency of about 1 in 40 residues makes it a promiscuous interactor (*Rosenzweig et al., 2017*), generating a highly heterogeneous Hsp70-substrate ensemble that ensures the efficient search of conformational space for an optimal folding pathway (*Lee et al., 2015*; *Rosenzweig et al., 2017*; *Sekhar et al., 2017*). The structural constraints of the Hsp70 binding pocket (*Figure 1C*) are such that only an unfolded stretch of polypeptide can fit into the binding groove if the helical lid is closed (*Zhu et al., 1996*), though more compact protein conformations can be accommodated if the lid is partially open (*Schlecht et al., 2011*).

Hsp70 function involves altering the conformation of client proteins (*Mayer, 2013*; *Rodriguez et al., 2008*; *Goloubinoff and De Los Rios, 2007*; *Clerico et al., 2015*) during folding (*Sharma et al., 2010*; *Szabo et al., 1994*) and disaggregation (*Mogk et al., 2015*), as well as dismantling of oligomeric assemblies such as clathrin coats (*Böcking et al., 2011*; *Sousa et al., 2016*) that surround vesicles involved in intra-cellular trafficking. However, the molecular mechanism by which Hsp70 causes these diverse conformational changes remains poorly understood. The structures of the Hsp70 chaperone in its ATP (*Kityk et al., 2012*; *Qi et al., 2013*) (*Figure 1D*) and ADP states (*Bertelsen et al., 2009*) (*Figure 1A*) are substantially different and the conformational switch that occurs in Hsp70 upon ATP hydrolysis has been hypothesized to perform conformational work

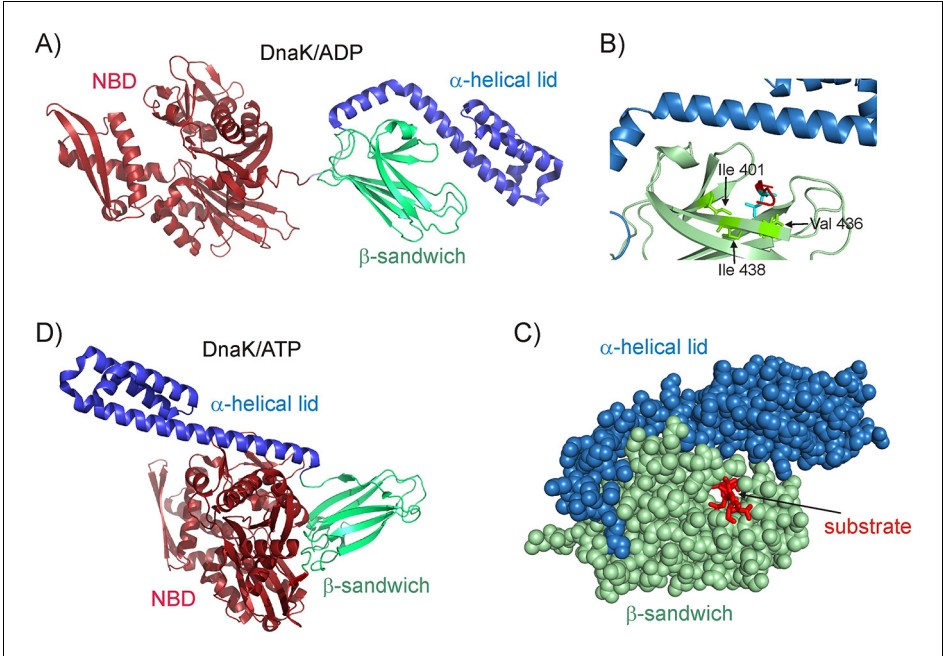

**Figure 1.** Structure of the *E.coli* Hsp70 (DnaK) chaperone. (**A**) Cartoon representation of the NMR structure of DnaK/ADP (*Bertelsen et al., 2009*) showing the N-terminal ATPase domain (NBD) and the C-terminal substrate binding domain which consists of a β-sandwich subdomain and an α-helical lid (PDB ID: 2KHO). (**B**) An enlarged view of the substrate binding cleft showing key hydrophobic residues Ile 401, Val 436 and Ile 438 of DnaK as green sticks forming contacts with the substrate residue that occupies the central binding position (cyan sticks) (PDB ID: 1DKZ) (*Zhu et al., 1996*). The remainder of the peptide substrate is shown in red. (**C**) Space filling structure of the substrate binding domain (SBD) with a closed conformation of the helical lid, illustrating the pore that the substrate (red) threads through. (**D**) The crystal structure of DnaK/ATP in the absence of substrate (*Kityk et al., 2012*; *Qi et al., 2013*), showing that upon ATP hydrolysis the α-helical lid becomes docked on the β-sandwich domain concomitant with the undocking of NBD and SBD (PDB ID: 4B9Q).
DOI: https://doi.org/10.7554/eLife.32764.003

on the bound substrate and alter its structure (*Mayer, 2013*; *Clerico et al., 2015*). Contrary to this hypothesis, NMR studies by our group have shown that the dominant native (*i.e.*, unbound) and bound conformations of a substrate can be significantly different despite the fact that the structure of the substrate in the ADP-bound, ATP-bound and nucleotide-free forms of Hsp70 are identical, showing that ATP hydrolysis does not have to be coupled to a substrate conformational change in the Hsp70 chaperone system (*Sekhar et al., 2015*).

Two limiting-case mechanisms have been proposed for the binding of Hsp70 to its target substrates, namely conformational selection (CS) and induced fit (IF), also referred to in some reports as the holdase and the unfoldase models, respectively (*Mayer, 2013*; *Sharma et al., 2010*; *Rodriguez et al., 2008*; *Goloubinoff and De Los Rios, 2007*; *Clerico et al., 2015*; *Böcking et al., 2011*; *Slepenkov and Witt, 2002*). In some cases conformational transitions in the substrate may occur in the absence of Hsp70 binding but still form an important component of the binding process. For example, the binding of Hsp70 to the clathrin triskelion that leads to downstream uncoating of a clathrin-coated vesicle has been postulated to occur via local conformational fluctuations in individual clathrin molecules that transiently expose the Hsp70 binding site (*Böcking et al., 2011*). The recognition of $\sigma^{32}$ by *E.coli* Hsp70 (DnaK) has also been hypothesized to occur through transient excursions of the $\sigma^{32}$ molecule from a compact conformation where the binding site is sequestered in a helix to a more accessible extended structure (*Rodriguez et al., 2008*). In contrast, the IF-like unfoldase model is believed to operate in accelerating the folding of large protein substrates such as luciferase by transforming misfolded kinetically trapped states into unfolded but folding-competent conformations (*Sharma et al., 2010*).

In general, distinguishing between CS and IF mechanisms (*Changeux and Edelstein, 2011*) requires a consideration of both structure and kinetics. For example, in order to establish that a particular binding interaction occurs via CS, it is necessary both to demonstrate that a very similar conformation to that in the bound state is sampled in the absence of the binder and to show that the formation of the complex occurs primarily via a kinetic pathway whereby the conformation of the ligand that resembles the bound form is selected for binding. While there is compelling evidence in a number of cases for the pre-sampling of a bound conformation in the unbound state (*Boehr et al., 2009*; *Tzeng and Kalodimos, 2009*; *Ye et al., 2016*; *Lange et al., 2008*; *Anthis et al., 2011*; *Venditti and Clore, 2012*), very few reports have unequivocally established that such a conformation is on-pathway to the bound state (*Birdsall et al., 1980*; *Daniels et al., 2014*; *Daniels et al., 2015*; *Chakrabarti et al., 2016*). Indeed, pre-sampling of a bound conformation does not automatically imply that the bound state forms from such a conformation. Several recent studies have sought to make this clarification by formulating the CS and IF mechanisms in terms of the flux of molecules flowing through the two pathways to the bound state (*Hammes et al., 2009*; *Weikl and Paul, 2014*).

Herein, in an effort to further characterize how Hsp70 chaperones bind to cognate substrates, we address which of the competing pathways, CS or IF, better explains the binding interaction between Hsp70 and its targets. We consider a pair of folding competent substrates with very different secondary structures (α-helix and β-sheet) and both *E. coli* and human (Hsc70) chaperones and probe the binding reaction using magnetization exchange (*Farrow et al., 1994*; *Palmer et al., 2001*) and chemical exchange saturation transfer (CEST) NMR methodology (*Vallurupalli et al., 2017*; *Anthis and Clore, 2015*), which simultaneously provide structural information about each of the participating components as well as the kinetics of the recognition event. The NMR experiments exploit selective isotope labeling (*Kerfah et al., 2015*) and methyl transverse relaxation optimized spectroscopy (methyl TROSY) (*Tugarinov et al., 2003*; *Ollerenshaw et al., 2003*) approaches so that large protein assemblies such as the Hsp70-substrate complex can be probed quantitatively at atomic resolution. We have carried out equilibrium measurements and analyzed the resulting kinetic parameters in terms of fluxes along the competing CS and IF pathways, where formation of the unfolded bound conformer is predominantly the result of chaperone binding to either unfolded or native substrate states, respectively. Our results establish that the dominant binding process is via the CS mechanism for both substrates considered, and for both DnaK and human Hsc70, suggesting that the CS mode of Hsp70 recognition may be conserved. Our data highlight the importance of molecular dynamics in Hsp70 chaperone-substrate binding and emphasize the potential of NMR spectroscopy in providing unambiguous mechanistic insights into key biological processes that involve molecular interactions.

## Results

### NMR experiments probing CS and IF modes of Hsp70-substrate interactions

In what follows we consider a solution comprising client protein substrate and Hsp70 (referred to as *K*) at equilibrium. The client protein is not characterized by a single structure, but rather by an ensemble of conformers, ranging from native (*N*) to unfolded (*U*); in the examples that follow both *N* and *U* states are approximately equally populated, and are in principle, both available to form a substrate-Hsp70 bound complex. A number of different biophysical experiments, performed on a variety of Hsp70 complexes, has established that the substrate is unfolded in the bound state, *UK* (*Palleros et al., 1994*; *Chen et al., 2006*; *Sharma et al., 2010*; *Kellner et al., 2014*; *Lee et al., 2015*; *Sekhar et al., 2015*). Our goal is to obtain mechanistic insights into the binding reaction by considering the flux from either conformers *U* or *N* to *UK* so as to evaluate what the dominant pathway for the Hsp70 - substrate interaction might be and hence establish whether binding is best described in terms of a CS or IF mechanism.

As described in the Introduction, in the IF (unfoldase) model the dominant process involves Hsp70 binding to *N* to form a preliminary complex (*NK*), where the bound substrate has an *N*-like structure, that is subsequently unfolded to form the dominant bound conformation (*UK*). The reaction can be denoted as $N \rightarrow NK \rightarrow UK$, although in principle more complex schemes can also be

included so long as they do not involve the $U \rightarrow UK$ step. In contrast, in the CS (holdase) model, as we consider it here, Hsp70 selects a substrate conformer that structurally resembles the bound state ($UK$) from an equilibrium mixture, with larger fluxes for reactions that proceed via this path ($U \rightarrow UK$) than for those involving $N \rightarrow NK \rightarrow UK$. It is important to emphasize that a number of pathways fit the description of the CS mechanism, including both $N \rightarrow U \rightarrow UK$ and the simpler $U \rightarrow UK$. Indeed, in cases where the $N,U$ conformational interconversion is limiting kinetically the flux through the $U \rightarrow UK$ pathway will be larger than the more complex CS scheme that begins with $N$.

Since the distinction between the CS and IF models involves a consideration of both structure and kinetics, our strategy here is to use magnetization exchange (zz-exchange) (*Farrow et al., 1994*; *Palmer et al., 2001*) or CEST NMR (*Vallurupalli et al., 2017*) experiments that both resolve different conformers through chemical shifts and facilitate extraction of rates of interconversion. *Figure 2A* shows a schematic of a zz-exchange spectrum for a two-site exchange reaction, $A \underset{k_{BA}}{\overset{k_{AB}}{\rightleftharpoons}} B$. In this experiment the chemical shifts of NMR signals are measured both before and after a delay during which chemical exchange occurs. Because the experiment is constructed essentially as a $^{13}$C-$^1$H HMQC, whereby $^{13}$C and $^1$H chemical shifts are recorded before and after the exchange period, respectively, a dataset is generated in which for each methyl group a pair of peaks (diagonal-peaks) is obtained at the resonance positions for the $^1$H and $^{13}$C methyl spins that are associated with the two interconverting states. An additional pair of peaks (cross-peaks) is also present that link the diagonal peaks, derived from the transfer of longitudinal magnetization from $A$ to $B$ or $B$ to $A$ during the exchange period. Quantification of the buildup and decay of the cross- and diagonal-peaks, respectively, as a function of the duration of the exchange period enables the extraction of rates of interconversion and equilibrium populations of states. *Figure 2B* illustrates a hypothetical CEST profile, again focusing on a two-state exchange process. In this experiment the intensity of an observed peak (say from a proton of residue $j$ of state A) is monitored as a function of the position of a weak radio frequency (rf) $^1$H field that perturbs magnetization resonating at or close to the frequency of application of the weak field. The field, applied for a duration $T_{EX}$, is scanned across all possible frequencies (one frequency per experiment), eventually perturbing the corresponding proton magnetization from residue $j$ of state B and this perturbation is subsequently transferred to the observed peak (A) due to chemical exchange. Thus, a plot of the intensity ratio of the A state peak ($I/I_o$, where $I$ and $I_o$ are the intensities of peak A when the field is ($T_{EX} \neq 0$) and is not ($T_{EX} = 0$) applied, respectively) as a function of frequency shows a major dip at the resonance position of the target proton in state A and a minor dip at the position of the resonance frequency of the proton in state B. Note that there is an analogy between the CEST profiles from A and B and a zz-exchange dataset, with the major dips in CEST corresponding to the diagonal peaks in zz-exchange, connected via the minor dips in the two profiles, similar to the zz-exchange crosspeaks. The resulting curves from zz-exchange or CEST profiles can be fit to the Bloch-McConnell equations (*McConnell, 1958*) that include the effects of chemical exchange to extract parameters of the interconversion process.

There are a number of important experimental constraints which must be taken into account in the choice of Hsp70 substrates for study via these two approaches. For example, if the $N,U$ interconversion is significantly faster than the binding step the distinction between $N$ and $U$ becomes 'blurred' and it becomes difficult to distinguish binding starting from $N$ or from $U$. This is illustrated in the context of a set of reactions,

$$N \underset{k_{UN}}{\overset{k_{NU}}{\rightleftharpoons}} U$$

$$U + K \underset{k_{off}}{\overset{k_{on}^{UK}}{\rightleftharpoons}} UK$$

where $N$ and $U$ are DnaK-free conformations of the substrate and $K$ refers to DnaK. Focusing on CEST, *Figure 2C* shows that the $N$ (red) and $U$ (green) profiles become indistinguishable for fast interconversion ($k_{ex,NU} \gg k_{ex,UK}$, where $k_{ex,NU} = k_{NU} + k_{UN}$, $k_{ex,UK} = k_{on}^{UK}[K] + k_{off}$; $k_{ex,NU} \ll \Delta\omega_{NU}$ where $\Delta\omega_{NU}$ is the chemical shift difference between corresponding peaks in states $N$ and $U$), while if the interconversion is slow ($k_{ex,NU} \ll k_{ex,UK}$) then the mechanism of binding can clearly be discerned from a comparison of the CEST profiles of $N$ and $U$, since only the latter shows a minor dip at the chemical shift of the reporter nucleus in the bound state in this case (*Figure 2D*). Similarly, assuming

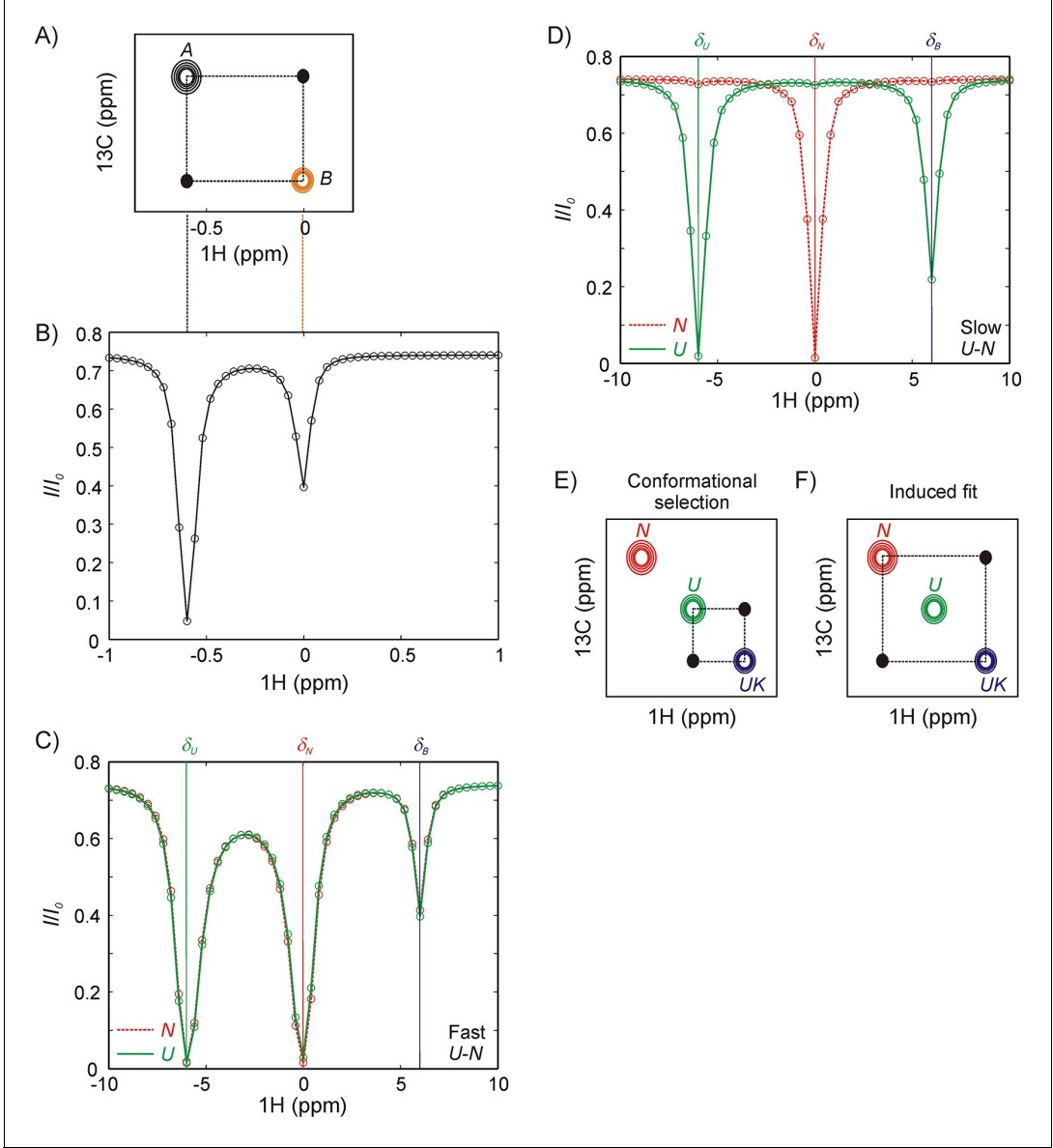

**Figure 2.** Magnetization transfer and CEST experiments for studies of binding kinetics. (A) Schematic of a zz-magnetization exchange spectrum showing diagonal peaks derived from magnetization in exchanging states A and B, $A \rightleftharpoons B$, as well as cross-peaks (black dots) connecting the two states. (B) Corresponding CEST profile for the exchange reaction in panel A that plots the normalized peak intensity of a spin in state A as a function of the offset at which a weak radiofrequency field is applied. A major dip at the chemical shift of the spin in state A ($-0.6$ ppm) as well as a minor dip at the corresponding chemical shift in state B (0 ppm), can be observed. Black and orange dashed lines between panels A and B show the correspondence between CEST and zz-exchange experiments via chemical shifts in states A and B. Note, however, that the CEST experiment amplifies the fingerprint of exchange, so that a minor dip for a spin in state B can be observed even if the peak cannot be directly visualized in the spectrum ($p_B$ <5%). NMR spin-relaxation methods can distinguish between IF and CS binding pathways so long as N and U conformations interconvert slowly (slower than the binding kinetics). If N and U interconvert rapidly compared to the kinetics of binding (C), any perturbation to the magnetization of the bound state by application of a weak radiofrequency field is rapidly conveyed to N even if binding is only to U, resulting in identical CEST profiles for N (red) and U (green). (D) In contrast, if states N and U interconvert slowly then distinct CEST profiles can be observed for CS and IF binding mechanisms, as shown here for binding to the U state via conformation selection. A prominent minor state dip is observed in the U profile (green) at the position of the bound frequency, $\delta_B$, corresponding to a flux through the $U \rightarrow UK$ reaction. Because a minor dip is not observed between N and UK (red) the flux through the reaction $N \rightarrow UK$ must be much smaller than via $U \rightarrow UK$. (E, F) Schematic of 2D planes derived from a $^{13}$C-$^1$H magnetization exchange experiment recorded on a system where the binding reaction is described by conformational selection, $U \rightarrow UK$ (E) or induced fit, $N \rightarrow UK$ (F). The U-N interconversion is assumed to be too slow to give rise to detectable cross-peaks in this experiment. Distinct cross-peaks are observed connecting the conformation that directly binds the partner (U in panel E, conformational selection and N in panel F, induced fit).

DOI: https://doi.org/10.7554/eLife.32764.004

$k_{ex,NU} \ll k_{ex,UK}$ in a zz-exchange experiment for the same CS-based mechanism as above (*Figure 2E*), cross-peaks are observed only between *U* and *UK*, while for an induced-fit mode of binding cross-peaks connect only *N* and *UK* (*Figure 2F*). In addition, rates of interconversion must be on the order of ~1 s$^{-1}$ or faster for zz-exchange (*Farrow et al., 1994*; *Palmer et al., 2001*) if all of the exchange parameters are to be extracted robustly, or ~20 s$^{-1}$ for the CEST experiment (*Vallurupalli et al., 2017*) in the general case where profiles from only one of the interconverting species can be measured. Notably, in cases where peaks from *N*, *U* and *UK* are visible in spectra this restriction is relaxed for the analysis of CEST profiles as we show below. Finally, it is worth emphasizing that, in the above discussion, we have assumed that exchange rates are slow compared to chemical shift differences between spins in interconverting states, *N*, *U* and *UK*, so that separate correlations are observed for each state in NMR datasets.

The *N* and *U* states defined above may themselves be composed of sub-ensembles of interconverting conformers, for example {$N_1,N_2,...$} and {$U_1,U_2,...$}. In constructing the CS and IF models in terms of *N* and *U* states it is assumed that the interconversion between sub-states $N_i$ or between $U_i$ occurs much faster than the *N-U* exchange or than ligand binding, with a rate that is fast on the NMR chemical shift timescale. In this manner only single distinct NMR peaks are observed for spins in *N* or *U* states rather than separate peaks for each element of the sub-ensembles ($N_i$) or ($U_i$) so that *N* or *U* can thus be treated as single 'averaged' conformers. It naturally follows that the affinities and rates obtained from NMR experiments will be averages over the members of each sub-ensemble. It is also the case that additional intermediates such as alternate unfolded states (*U**) can be explicitly included in any model of exchange only if their presence is detected either by distinct peaks in NMR spectra or by minor dips in CEST profiles.

## Choice of substrates and chaperones

As a first substrate we focused on the slow-folding marginally stable L90A mutant of the 17th domain of chicken-brain α-spectrin (referred to as R17*) (*Scott et al., 2006*). R17* is a three-helix bundle protein domain (*Figure 3A*) with kinetics and thermodynamics of folding that suggest that it may be an ideal candidate for distinguishing between different Hsp70 binding mechanisms using the NMR approaches described above (*Scott et al., 2006*). R17* has a strong predicted (*Rüdiger et al., 1997*) Hsp70 binding site and the sidechains ($^{41}$IQGLL$^{45}$) forming this binding site are >40% solvent-exposed on average (*Figure 3B*), potentially facilitating binding of Hsp70 to the native protein along an IF pathway. On the other hand the low stability of the folded domain suggests a large population of the *U* state in solution (see below) that might favour the CS route.

*E. coli* DnaK is the Hsp70 ortholog that has been most extensively characterized at structural (*Kityk et al., 2012*; *Qi et al., 2013*; *Bertelsen et al., 2009*; *Pellecchia et al., 2000*; *Harrison et al., 1997*), kinetic (*Sharma et al., 2010*; *Slepenkov and Witt, 2002*; *Gisler et al., 1998*; *Grimshaw et al., 2001*; *Grimshaw et al., 2003*; *Han and Christen, 2003*; *Pierpaoli et al., 1998*; *Schmid et al., 1994*; *Chesnokova and Witt, 2005*; *Slepenkov and Witt, 1998*; *Jordan and McMacken, 1995*; *Russell et al., 1998*) and mechanistic levels (*Sharma et al., 2010*; *Szabo et al., 1994*; *Landry et al., 1992*; *Schröder et al., 1993*; *Schönfeld et al., 1995*; *Greene et al., 1998*; *Montgomery et al., 1999*; *Swain and Gierasch, 2006*; *Swain et al., 2007*; *Smock et al., 2010*; *Zhuravleva and Gierasch, 2015*; *Zhuravleva et al., 2012*; *Moro et al., 2003*; *Moro et al., 2004*; *Taneva et al., 2010*; *Mayer et al., 2000*; *Laufen et al., 1999*), and it was therefore selected for initial experiments. The ATP state of DnaK is an important entry point for substrates to the Hsp70 chaperone cycle (*Pierpaoli et al., 1998*) and was chosen as the chaperone nucleotide state in this study. In the ATP state DnaK substrate complexes have millisecond lifetimes and have been previously studied with CEST methodology (*Sekhar et al., 2015*). In all experiments involving DnaK/ATP, the allosterically active T199A mutant of DnaK (*McCarty and Walker, 1991*) that is deficient in ATP hydrolysis was used to minimize ATP turnover during NMR data collection.

## R17* binds DnaK

*Figure 3C* shows the Met methyl region of a $^{13}$C-$^{1}$H HMQC spectrum of 250 µM U-$^{2}$H, Ileδ1-$^{13}$CH$_3$, Leu/Val-$^{13}$CH$_3$/$^{12}$CD$_3$, Met-$^{13}$CH$_3$ (ILVM-$^{13}$CH$_3$) labeled R17*, 25°C. There are two Met residues in R17* and five resonances in the spectral region. Three of the peaks were assigned to Met 87 and two to Met 26 based on the $^{13}$C-$^{1}$H HMQC spectrum of M26L R17* (*Figure 3—figure supplement*

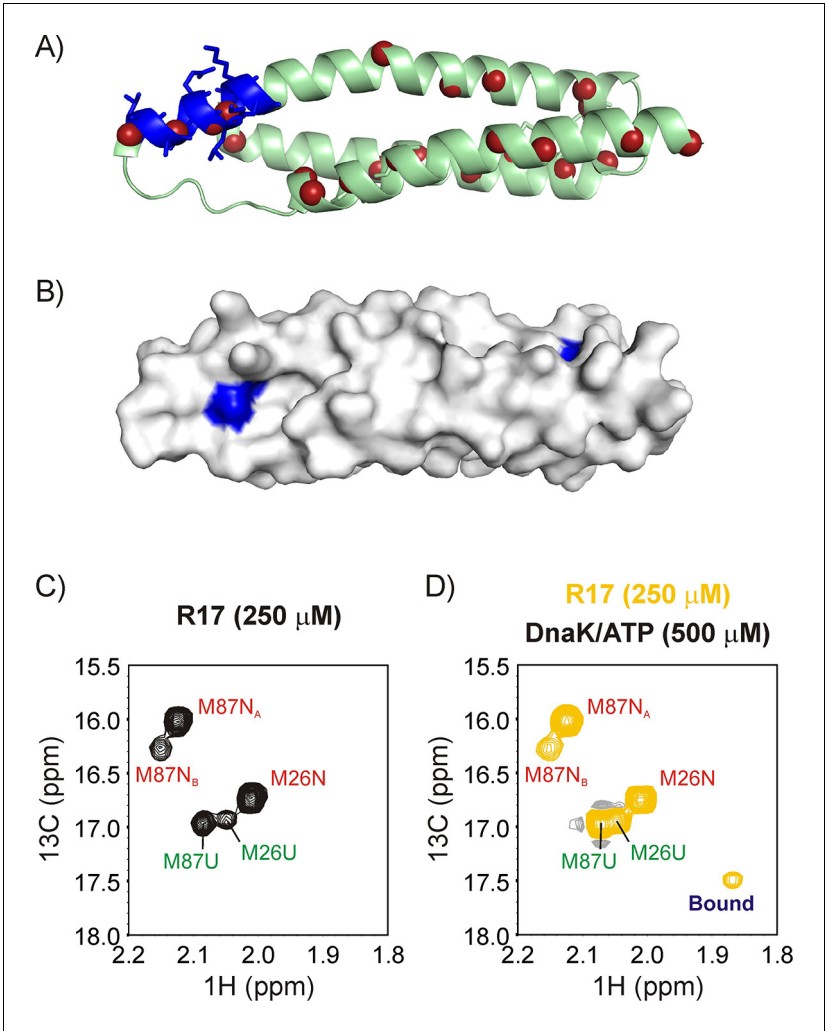

**Figure 3.** R17* binds DnaK. Ribbon diagram of the X-ray structure of R17 (PDB ID: 1CUN) (*Grum et al., 1999*) showing the strongest DnaK binding site in blue with sidechains denoted as sticks. The backbone nitrogen atoms of methyl containing amino acids Ile, Leu, Met and Val are indicated as red spheres; these methyls are used as probes of chaperone binding with the labeling scheme that we employ. (**B**) Surface representation of the R17 structure, with the Ile and Leu residues of the two DnaK binding sites identified from LIMBO (*Van Durme et al., 2009*) shown in blue. Met region of the $^{13}C$-$^{1}H$ HMQC spectrum recorded on a 250 µM ILVM $^{13}CH_3$-labeled R17* sample without (**C**) and with (**D**) 500 µM DnaK/ATP, 25°C. Assignments of folded (N) and unfolded (U) Met are indicated in red and green, respectively. The peak that appears upon DnaK addition is labeled as 'Bound'.

DOI: https://doi.org/10.7554/eLife.32764.005

The following figure supplements are available for figure 3:

**Figure supplement 1.** Resonance assignments for R17*.

DOI: https://doi.org/10.7554/eLife.32764.006

**Figure supplement 2.** Identifying residues of R17* at the central binding groove of DnaK.

DOI: https://doi.org/10.7554/eLife.32764.007

---

*1A*). In addition, we carried out a temperature titration of R17* to gradually unfold the protein so as to ascertain whether the extra peaks arise from its unfolded state. *Figure 3—figure supplement 1B* shows that only two of the resonances remain in a $^{13}C$-$^{1}H$ HMQC spectrum recorded at 45°C and these are therefore assigned to the unfolded state of R17. The other three resonances derive from the native conformer, with the weaker resonance from Met 87 (M87N$_B$) possibly originating from an R17* dimer that is detected on the size exclusion column during purification.

Upon addition of 500 μM U-$^2$H DnaK/ATP a new peak appears in the Met region of the $^{13}$C-$^1$H HMQC spectrum that can be unambiguously assigned to the bound state of R17* (*Figure 3D*). The binding of R17* to DnaK can also be established from the $^{13}$C-$^1$H HMQC spectrum of a sample prepared as 400 μM ILVM-$^{13}$CH$_3$ R17* and 800 μM I-$^{13}$CH$_3$ DnaK/ADP in which a number of new peaks are observed for the key Ile 401 and Ile 438 residues present at the central binding pocket of DnaK that are sensitive reporters of the binding event (*Rosenzweig et al., 2017*) (*Figure 3—figure supplement 2*, middle). The presence of several Ile 401 and Ile 438 resonances at distinct chemical shifts in the bound state demonstrates the existence of multiple bound conformations of the R17*-DnaK complex, interconverting with each other and with free DnaK in slow exchange on the NMR chemical shift timescale. Such conformational heterogeneity in DnaK-substrate interactions, observed previously in studies focused on the three-helix bundle client hTRF1 (*Rosenzweig et al., 2017*), arises from the binding of DnaK to a number of sites on R17* that position different aliphatic hydrophobic amino acids of the client at the central groove of the DnaK binding site (referred to as the 0 position). Indeed, the DnaK binding site prediction algorithm LIMBO (*Van Durme et al., 2009*) identifies at least two regions on R17* with a high propensity to interact with DnaK, $^{42}$QGLLKKH$^{48}$ and $^{70}$EDLIKKN$^{76}$ (*Figure 3B*), each of which has two hydrophobic residues (L44 and L45 in region 1 and L72 and I73 in region 2) that could occupy the central cavity. Notably, substitution of the Leu pair at positions 44 and 45 with Ile leads to the appearance of new correlations in the Ile region of a $^{13}$C-$^1$H HMQC spectrum recorded on a 200 μM ILVM-$^{13}$CH$_3$ L44I/L45I R17* sample to which 400 μM $^2$H DnaK/ADP has been added, confirming that the region containing residues 42–48 is a DnaK binding site. We have not been able to confirm DnaK binding to the second predicted site, as substitution of Leu 72 with Ile did not yield additional correlations in the Ile region of a spectrum recorded on a suitably prepared R17*, DnaK/ADP sample.

In order to determine whether the new resonance in the HMQC spectrum of the R17*-DnaK complex belongs to a Met residue at the 0 position of the DnaK binding site we measured intermolecular NOEs between ILVM-$^{13}$CH$_3$ R17* and I-$^{13}$CH$_3$ DnaK/ADP (*Figure 3—figure supplement 2*). Here the ADP state of DnaK is used as it has higher affinity for substrates and the complex has lower $k_{off}$ rates (*Mayer and Bukau, 2005*). Previous X-ray and NMR structures of a variety of different complexes have established that Ile 401 and Ile 438 in the DnaK binding pocket are within 5 Å of the methyl moiety of the substrate residue located at position 0 (*Zhu et al., 1996*; *Zahn et al., 2013*; *Stevens et al., 2003*) and NOEs have been used to link Ile 401, Ile 438 cross peaks with those from substrate residues that belong to the same conformation (*Rosenzweig et al., 2017*). For R17* the bound Met methyl does not show NOE cross-peaks to Ile 401 and Ile 438 of DnaK, establishing that this Met residue is not located at the 0 position (*Figure 3—figure supplement 2*, left). However, NOE cross-peaks can be observed connecting pairs of Ile 401 and Ile 438 residues (labeled as 'a' and 'b') with their substrate counterparts whose methyls are localized to the Leu region of the HMQC spectrum, establishing that at least two distinct Leu residues of R17* can be present at the DnaK 0 position in the R17*-DnaK ensemble (*Figure 3—figure supplement 2*, right).

A number of biophysical studies using different spectroscopies such as NMR, fluorescence and circular dichroism have shown that DnaK-bound substrates are globally unfolded (*Palleros et al., 1994*; *Chen et al., 2006*; *Sharma et al., 2010*; *Kellner et al., 2014*; *Lee et al., 2015*; *Sekhar et al., 2015*). In order to determine the conformation of R17* in its DnaK-bound state we have compared the intensities of cross-peaks in $^{13}$C-$^1$H HMQC spectra derived from native and unfolded states of R17* as probes. The intensities of native state Met and Ile resonances decrease by a factor of 1.6 ± 0.1 upon DnaK binding, consistent with solution equilibria shifting from *N*, and also with different $^1$H and $^{13}$C chemical shifts of R17* in the bound state from those in *N*. In contrast, intensities of Ile and Met correlations derived from the unfolded state increase by 2.2 ± 0.6 fold. The simplest interpretation of the data is that the Met and Ile residues of R17* in bound conformations have the same chemical shifts as *U*, with the degeneracy in shifts leading to an increase, rather than a decrease in unfolded peak intensities. This provides strong evidence that R17* is globally unfolded in the DnaK-bound state. The distinct $^{13}$C and $^1$H chemical shifts from the Met peak labeled 'Bound' in *Figure 3D* (assigned as Met 26, see below), that are not unfolded-like, reflect a conformation of R17*-DnaK whereby the Met is proximal to the DnaK binding site and hence shifted from the random coil region of the spectrum. Taken together, our data shows that DnaK-bound R17* is globally unfolded and resembles the *U* conformation of R17* that exists in equilibrium with *N* in the absence of DnaK.

## Quantifying the mechanism of DnaK binding to R17*

Having established that the bound form of R17* is structurally very similar to $U$ we next turned to studying the binding reaction using zz-exchange magnetization transfer. This experiment is particularly powerful in the case of R17* because both the $^1$H and $^{13}$C chemical shifts of the one Met resonance that reports on the bound state are distinct from those derived from $N$ and $U$, and the Met region of the $^{13}$C-$^1$H HMQC spectrum is well resolved, facilitating the detection and quantification of cross-peaks between states. *Figure 4A* shows a 2D plane from the zz-exchange dataset acquired with a mixing time of 300 ms. A pair of cross-peaks can be observed, labeled 1 and 2, connecting the unfolded state of Met 26 with the bound state and assigning the bound resonance to the methyl group of Met 26. Notably, there are no cross-peaks between $N$ and the bound state, unequivocally demonstrating that DnaK interacts predominantly with the unfolded state of R17*.

In order to obtain the binding and release rate constants for the reaction, $U \underset{k_{BU}}{\overset{k_{UB}}{\rightleftharpoons}} UK$, we measured diagonal (unfolded and bound) and cross-peak intensities for Met 26 as a function of mixing time values that ranged from 25 to 800 ms and fit the resulting profiles globally to extract $k_{UB}$ and $k_{BU}$ values of $0.8 \pm 0.1$ s$^{-1}$ and $2.7 \pm 0.2$ s$^{-1}$, respectively (*Figure 4B*). One dimensional reduced $\chi^2$ surfaces show that both rate constants can be obtained reliably from the exchange dataset (*Figure 4C*), with errors in the pseudo-first order rate constant $k_{UB}$ and the first order constant $k_{BU}$ estimated by a bootstrapping procedure (*Efron and Tibshirani, 1986*), described in Materials and methods (*Figure 4D*). On the basis of the relative fractional populations of $N$ and $U$ ($p_N/p_U$), as estimated from the first plane of the pseudo-3D zz-exchange dataset with $T_{MIX}$ = 25 ms (1.8:1) along with $p_U/p_B$ (=$k_{BU}/k_{UB}$) values from the time-dependent zz-exchange data (3.4:1) $p_N$, $p_U$ and $p_B$ (bound fraction) are calculated to be $58 \pm 2\%$, $33 \pm 1\%$ and $9 \pm 2\%$, respectively. From the total DnaK concentration of 500 μM in the sample the bimolecular rate constant for the binding reaction is calculated to be $k_{on}^{UB} = k_{UB}/[DnaK]$=1600 ± 250 M$^{-1}$s$^{-1}$. Note that this value is an underestimate because there are several bound conformations in solution (see discussion above) and we only focus here on the process for which the cross-peak of Met 26 of the bound state is resolved. Other complexes for which the Met 26 cross-peaks of the bound state are degenerate with the unfolded state are invisible to this analysis.

Because there are no cross-peaks between states $N$ and $B$ only upper-estimates of $k_{NB}$ and $k_{BN}$ rates can be obtained by assuming that the $N$-$B$ cross-peaks are at the noise level. Computations were performed, following the description outlined previously (*Huang et al., 2016*), where relative intensities of the $N$ and $B$ diagonal-peaks and the $N$-$B$ cross-peaks are calculated using experimentally measured values of relaxation times for longitudinal order, as a function of $k_{NB}$ and $k_{BN}$. Upper bounds for $k_{UN}$ and $k_{NU}$ were calculated in a similar manner as for $k_{NB}, k_{BN}$ since exchange-based cross-peaks reporting on the $U,N$ interconversion are absent as well. The values for all of the rate constants (including upper bounds in some cases) are illustrated in *Figure 4E* that shows the R17* binding mechanism.

The flux along the CS pathway is theoretically defined as the sum of fluxes for all pathways leading to the bound state via $U$ that do not involve the $NK \rightarrow UK$ transition. Similarly, the net flux through the IF pathway is the sum of the flux values for all paths that do not involve the $U \rightarrow UK$ step. In practice, however, the $U \leftrightarrow N$ transition is very slow for R17* with an upper bound of 0.01 s$^{-1}$ for $k_{ex,UN}$ so that the dominant flux contribution along the CS pathway derives from the $U \rightarrow UK$ transition while an upper bound for the IF pathway ($N \rightarrow NK \rightarrow UK$) can be estimated from the absence of an $N,UK$ cross-peak. The relative partitioning of the flux along the CS ($F_{CS}$) and IF ($F_{IF}$) paths, $\Theta$, is thus defined as

$$\Theta = \frac{F_{CS}}{F_{IF}} = \frac{k_{on}^{UB}[DnaK][U]}{k_{on}^{NB}[DnaK][N]} = \frac{k_{UB}\,p_U}{k_{NB}\,p_N}$$

From $k_{UB}$ = 0.8 s$^{-1}$ and $k_{NB}$ <0.004 s$^{-1}$, as well as $p_N/p_U$ = 1.8, $\Theta$ >113. Flux measurements thus confirm that the mechanism of R17* binding to DnaK is overwhelmingly biased in favour of CS (*Figure 4E*). A histogram of the quantified flux values for the $U \rightarrow UK$ (lower bound), $N \rightarrow U \rightarrow UK$ and $N \rightarrow NK \rightarrow UK$ pathways (only upper bounds) is presented in *Figure 4F*.

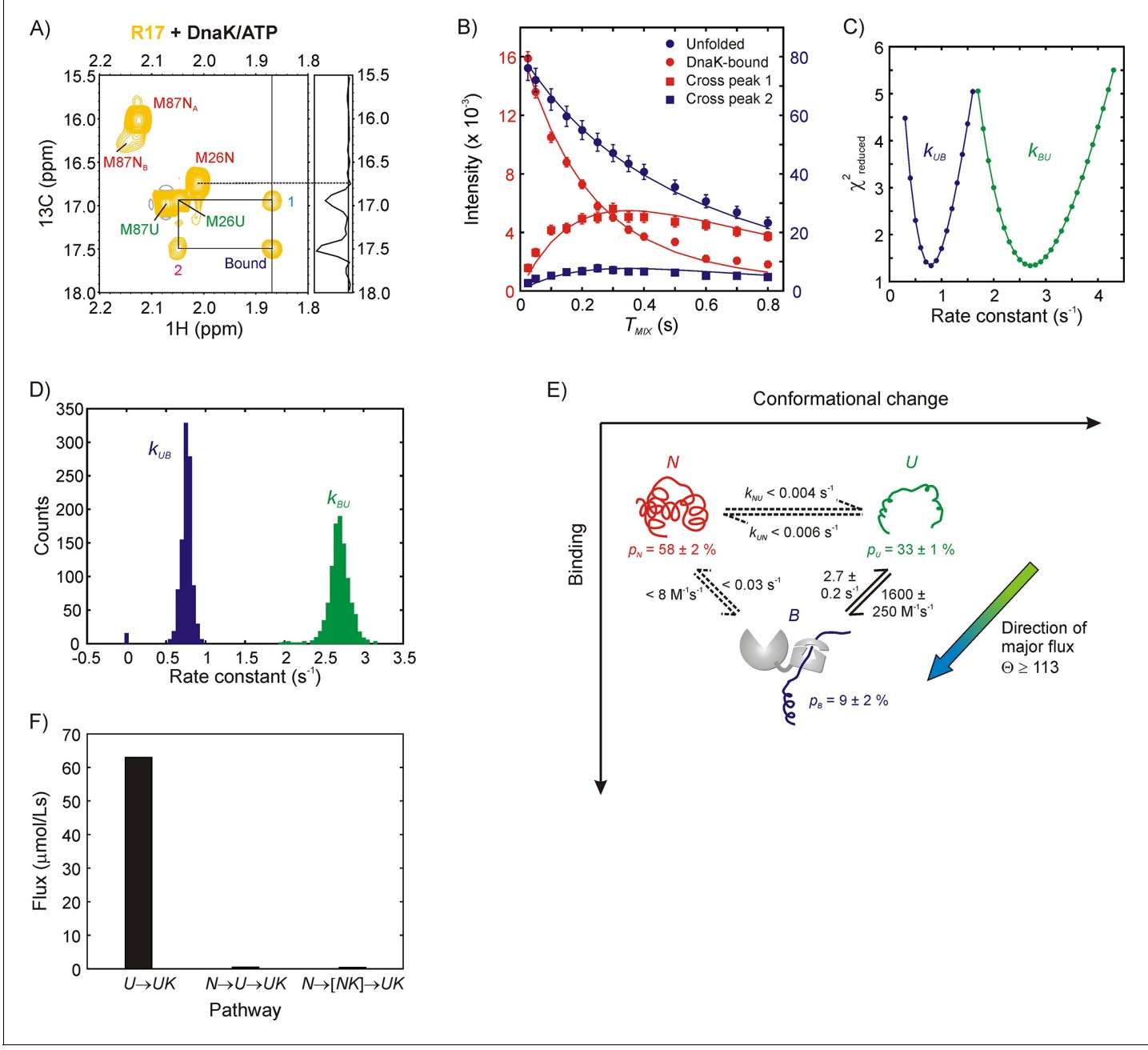

**Figure 4.** Mechanism of R17* binding to DnaK/ATP. (**A**) Magnetization exchange spectrum recorded on a 250 μM ILVM $^{13}CH_3$-labeled R17* sample (500 μM DnaK/ATP), focusing on the Met region. A mixing time of 300 ms was used so that exchange cross-peaks 1 and 2 could be easily observed. The exchange peaks establish the identity of the resonance labeled 'Bound' - it originates from Met 26. The 1D trace at the chemical shift denoted by the vertical solid line at 1.87 ppm is shown alongside the spectrum. Importantly, both the spectrum and the 1D trace show that cross-peaks are observed only from the bound state to the unfolded state and not to the native state of Met 26. (**B**) Mixing time-dependent changes in cross- (square) and diagonal- (circle) peak intensities of Met 26. The unfolded diagonal peak and the bound-to-unfolded cross-peak (cross-peak 2 in panel A) are coloured blue and matched with the blue axis labels (right) while the bound diagonal peak and the unfolded-to-bound cross-peak (cross-peak 1 in panel A) are shown in red along with the red axis labels (left). Solid curves are fits of the intensities to a two-state model of conformational exchange. (**C**) Dependence of reduced $\chi^2$ values from fits of the exchange data as a function of the pseudo first order rate constant $k_{UB}$ (blue) or the first order rate constant $k_{BU}$ (green). Values increase steeply from the minimum indicating that both rate constants can be obtained reliably from modeling the zz-exchange data. (**D**) Distributions in $k_{UB}$ and $k_{BU}$ based on a bootstrap analysis of the data, as described in Materials and methods; means and standard deviations of the distributions were used as parameter estimates and their errors, respectively. (**E**) Model of R17* binding to DnaK. Reactions for which only upper limits for the rate constants can be obtained are shown as dotted lines. The major flux for R17*-DnaK binding occurs along the elementary

*Figure 4 continued on next page*

*Figure 4 continued*

conformational selection pathway $U$ to $UK$. (**F**) Histogram showing flux values for the $U \rightarrow UK$, $N \rightarrow U \rightarrow UK$ (upper bound) and $N \rightarrow NK \rightarrow UK$ (upper bound) pathways. Note that $NK$ is not directly detected in our experiments.

DOI: https://doi.org/10.7554/eLife.32764.008

## Binding of a β-sheet substrate, drkN SH3

Having established that the binding of the α-helical bundle domain R17* to DnaK proceeds through a CS mechanism we next addressed the dependence of binding on the secondary structure of the client protein by choosing the incomplete 5-stranded β-barrel SH3 domain of the *Drosophila melanogaster* Enhancer of sevenless 2B protein (drkN SH3, *Figure 5A,B*) as a client. DrkN SH3 is a slow-folding protein with $k_{ex,UN} \sim 1$ s$^{-1}$, 20°C, and is marginally stable, populating a significant fraction of $U$ at equilibrium ($p_N/p_U \sim 2$, pH 6.0, 20°C) (*Farrow et al., 1994*; *Tollinger et al., 2001*). Previous studies have established that the DnaK bound state of drkN SH3 is structurally similar to $U$ (*Lee et al., 2015*).

*Figure 5* shows the methyl spectrum of 250 μM IM-$^{13}$CH$_3$ drkN SH3 without (**C**), and with (**D**) 500 μM U-$^2$H DnaK/ATP. The binding of SH3 to DnaK can be clearly visualized through the appearance of at least four new peaks, labeled 1–4 in *Figure 5D*. Two of these, 1 and 2, are particularly intense and separated in the $^1$H dimension from other resonances derived from the $N$ and $U$ states of drkN SH3. The $^{13}$C-$^1$H HMQC spectrum of a sample of 500 μM IM-$^{13}$CH$_3$ DnaK/ADP and 250 μM ILVM-$^{13}$CH$_3$ drkN SH3 is shown in *Figure 5—figure supplement 1* (middle). There are at least three new resonances for each of the DnaK binding site residues Ile 401 and Ile 438 when drkN SH3 is present, confirming that the SH3 domain binds to the canonical binding pocket on DnaK and highlighting the conformational heterogeneity in the SH3-DnaK ensemble, as has been seen in all other folding competent clients that we have studied to date. The identity of the residues of drkN SH3 at position 0 of the DnaK binding pocket have been obtained by recording a 3D $^{13}$C-edited NOESY dataset on the same sample (*Figure 5—figure supplement 1*). Notably, there are NOEs connecting the two Ile peaks 1 and 2 of drkN SH3 to Ile 438 of the chaperone, indicating that peaks 1 and 2 arise from Ile at the 0 position of the DnaK binding site. In contrast, NOEs from Leu residues in drkN SH3 to Ile 401 and Ile 438 are not observed. The absence of Leu residues at the binding site may possibly account for the low affinity of drkN SH3 for DnaK, even in the high affinity binding state where DnaK is ADP loaded. Cavagnero and coworkers report a $K_D$ of 243 μM for drkN SH3 and DnaK/ADP (*Lee et al., 2015*) that is at least two orders of magnitude weaker than literature values for peptides or proteins where Leu residues can occupy position 0 (*Pierpaoli et al., 1998*).

## Quantifying the mechanism of DnaK binding to drkN SH3

While the $^1$H chemical shifts of the bound Ile peaks 1 and 2 from drkN SH3 are unique, their $^{13}$C shifts are nearly degenerate with those from Ile 27 and Ile 4 of the $N$ state, respectively, (*Figure 5D*) which complicates the quantification of potential $N$-$B$ cross-peaks in a zz-exchange dataset. Thus, we have measured $^1$H CEST spectra to obtain the kinetics of drkN SH3 binding to DnaK/ATP using the pulse sequence shown in *Figure 6—figure supplement 1A*. Our choice of exploiting the $^1$H nucleus in CEST studies is motivated by the fact that the largest chemical shift differences between folded, unfolded and bound state peaks in methyl-based $^{13}$C-$^1$H HMQC datasets of drkN SH3 are in the $^1$H dimension. However, $^1$H CEST can be prone to artifacts arising from NOE effects that lead to spurious minor dips that do not report on conformational exchange (*Sekhar et al., 2016*; *Bouvignies and Kay, 2012*). In order to minimize these NOE-dips we have used highly deuterated I-$^{13}$CH$_3$ labeled drkN SH3 for acquiring CEST data and fortunately, there are only two pairs of Ile residues that are within 5 Å, considerably reducing the number of possible NOE dips. Finally, the mixing time during which exchange is quantified was set to 150 ms, a reasonably small value that further minimizes the sizes of these spurious dips. In principle it is possible to eliminate NOE effects completely using a spin-state selective $^1$H CEST scheme that has been proposed recently (*Yuwen et al., 2017*). However, this comes at a considerable cost in sensitivity (more than a factor of two) and since signal-to-noise is limiting in applications involving very slowly exchanging conformers (see below) or when short mixing times are selected we have chosen not to use this approach here.

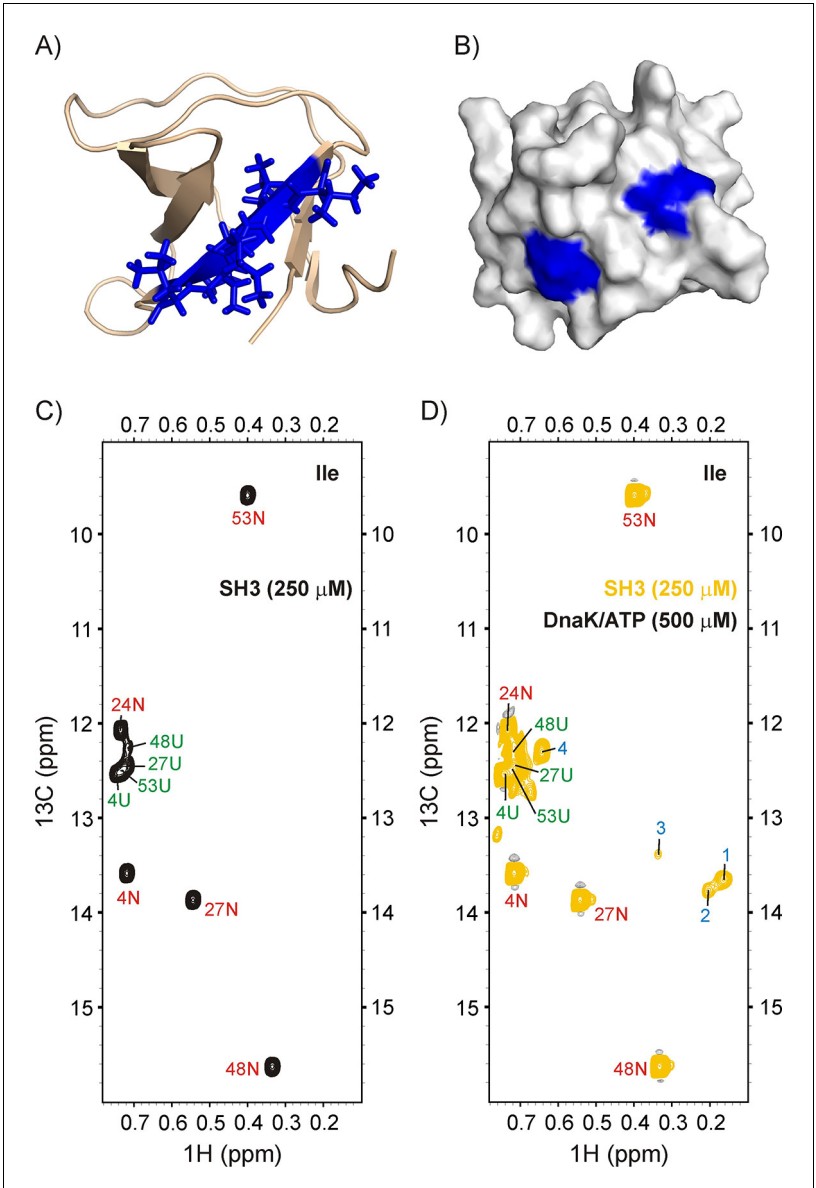

**Figure 5.** Binding of drkN SH3 to DnaK. (**A**) Cartoon representation of the structure of drkN SH3 (PDB ID: 2A36) (*Bezsonova et al., 2005*) with the strongest predicted DnaK binding site coloured in blue. (**B**) Surface representation of drkN SH3 where the large aliphatic hydrophobic sidechains of the two DnaK binding sites identified by the algorithm LIMBO (*Van Durme et al., 2009*) are shown in blue, illustrating that the binding sites are partially accessible even in the *N* state. Ile regions of $^{13}C$-$^{1}H$ HMQC spectra of 250 µM IM-$^{13}CH_3$ drkN SH3 without (**C**) and with (**D**) 500 µM U-$^{2}H$ DnaK/ATP, 25°C. Ile cross-peaks derived from *N* and *U* are labeled in red and green, respectively, while DnaK-bound peaks 1 to 4 that appear only upon DnaK addition are denoted by blue labels.

DOI: https://doi.org/10.7554/eLife.32764.009

The following figure supplement is available for figure 5:

**Figure supplement 1.** $^{13}C$-$^{1}H$ HMQC spectrum of 250 µM ILVM-$^{13}CH_3$ drkN SH3 containing 500 µM IM-$^{13}CH_3$ DnaK/ADP (middle).

DOI: https://doi.org/10.7554/eLife.32764.010

In order to interpret CEST profiles it was necessary to first assign peaks 1 and 2 to specific Ile residues in drkN SH3. This was achieved using a zz-exchange experiment that links peaks 1 and 2 to the corresponding *U* state correlations, that were already assigned by traditional NMR methods. In this way we could show that peaks 1 and 2 are derived from Ile 53 and Ile 27, respectively (*Figure 6—figure supplement 2*). Since we know from NOE data that both peaks 1 and 2 are present at the 0 position, it follows that peaks 1 and 2 report on two distinct DnaK-bound conformations of drkN SH3, labeled as B1 and B2 respectively. In the B1 conformer Ile 53 is at the 0 position while in B2 the 0 position is occupied by Ile 27. *Figure 6* shows $^1$H CEST profiles from Ile 27 and Ile 53 methyl groups of folded (A,D), unfolded (B,E) and DnaK-bound (C,F) drkN SH3, acquired on a sample with 250 µM IM-$^{13}$CH$_3$ drkN SH3 and 500 µM U-$^2$H-DnaK/ATP. Recall that in CEST spectra measured by recording the intensity of a peak from state *j* as a function of irradiation frequency the major dip derives from that state, while minor dips link state *j* with other states in the exchanging network. The native state CEST profiles of both Ile 27 and Ile 53 show only one minor dip each, at the resonances frequencies of the corresponding $^1$H$^{\delta 1}$ shifts in their respective unfolded states. These minor dips result from the *U-N* conformational exchange. However, the corresponding CEST traces from both residues in *U* show minor dips to *N* as well as to the bound state, unambiguously demonstrating that the major flux for the binding reaction involves a pathway whereby DnaK selects the unfolded state of drkN SH3. This conclusion is reinforced by CEST profiles of the bound state peaks where only one strong minor dip is observed in each trace that links the bound state with *U*. In contrast to DnaK/ATP, the CEST profiles of drkN SH3 bound to DnaK/ADP do not show any signature of binding/release (*Figure 6—figure supplement 1B*), though the presence of the bound states B1 and B2 can clearly be discerned from $^{13}$C-$^1$H HMQC spectra recorded on the same sample. This is a consequence of the slow rates of substrate release from DnaK/ADP ($\sim$0.001 s$^{-1}$, 25°C) (*Mayer and Bukau, 2005*; *Pierpaoli et al., 1998*), which places the exchange process outside the window of detection via CEST methods (*Vallurupalli et al., 2017*). The presence of CEST minor dips with DnaK/ATP and not with DnaK/ADP also confirms that DnaK is predominantly in the ATP-bound form in the sample used for acquiring CEST data in *Figure 6*.

Analysis of the $^1$H CEST profiles for the drkN SH3 system is challenging because of the measurable exchange between *U* and *N*, and *U* and *UK*, and by the presence of two bound states. Accordingly, we fit the CEST profiles from the Ile 27 and Ile 53 *N* and *U* peaks, as well as the profiles of I53B1 and I27B2 peaks globally to the four-state model shown in *Figure 7*. The intermediate state along the IF pathway, corresponding to *NK*, is not included in the model because there is no evidence from $^{13}$C-$^1$H HMQC spectra of SH3/DnaK samples or from the CEST profiles justifying the explicit inclusion of such a state; instead, the three-state *N-NK-UK* pathway was approximated as *N-UK*. The rate constants and populations that have been obtained from our modeling analysis are shown in *Figure 7*. The reliability of the kinetics parameters was evaluated by generating 1D reduced $\chi^2$ surfaces as a function of each of the eight global variables (5 rates, 3 populations; *Figure 7—figure supplement 1*). These show pronounced minima for all five rate constants $k_{ex,UB1}$, $k_{ex,NB1}$, $k_{ex,UB2}$, $k_{ex,NB2}$ and $k_{ex,UN}$. However, the minima are very shallow and broad for the populations, indicating that only the rate constants and not the populations can be obtained reliably from the CEST data.

The flux ratio $\Theta$ for the *U→UK* and *N→UK* pathways based on the simultaneous fit of CEST profiles from Ile 27 and Ile 53 is given by

$$\Theta = \frac{k_{on}^{UB}[DnaK][U]}{k_{on}^{NB}[DnaK][N]} = \frac{k_{ex,UB}\left(\frac{p_B}{p_U+p_B}\right)p_U}{k_{ex,NB}\left(\frac{p_B}{p_N+p_B}\right)p_N} = \frac{k_{ex,UB}}{k_{ex,NB}}\Xi$$

where $k_{ex,UB}$ and $k_{ex,NB}$ for both the B1 and B2 arms of the binding scheme of *Figure 7* can be obtained reliably. In order to estimate $\Xi$, we fit the CEST data using a bootstrap procedure (*Efron and Tibshirani, 1986*) described in Materials and methods; the resulting distribution of $\Xi$ values is narrow and centred around $\sim$1, with 98% of the values falling within the range of 0.84–1.27 for state B1 (Ile 53 at the 0 position) and 0.49–1.57 for B2 (Ile 27 at the 0 position) (*Figure 7—figure supplement 2*). The average and standard deviation of the $\Xi$ distribution are 1.0 ± 0.1 and 1.0 ± 0.2 for the B1 and B2 arms respectively. This implies that $\Theta$ is well approximated by the ratio $k_{ex,UB}/k_{ex,NB}$. Rate constants and their errors, as reported in *Figure 7*, are listed as the means and standard deviations of the respective distributions obtained from a bootstrapping analysis. Using these values, $\Theta 1$ and $\Theta 2$ for the formation of bound states

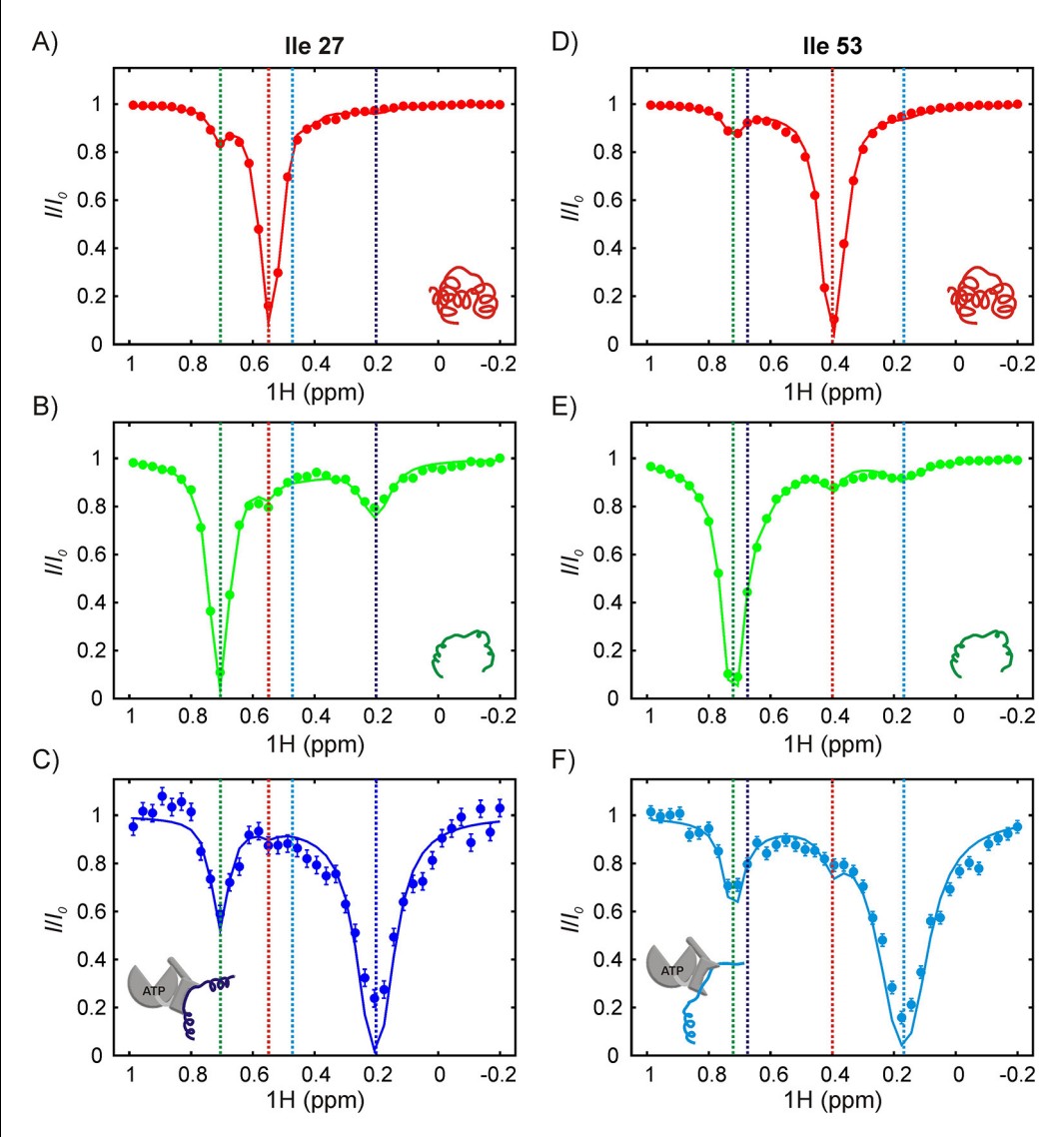

**Figure 6.** The SH3-DnaK binding interaction as detected by $^1$H CEST. $^1$H CEST profiles quantified from Ile 27 (A–C) and Ile 53 (D–F) native (A,D), unfolded (B,E) and DnaK-bound (C,F) resonances are highlighted. Measurements were performed on a 250 µM IM-$^{13}$CH$_3$ drkN SH3 sample containing 500 µM U-$^2$H DnaK/ATP. Vertical dotted lines highlight $^1$H$^{\delta 1}$ chemical shifts of Ile 27 (A–C) or Ile 53 (D–F) in the following conformations: native (red), unfolded (green), bound with Ile 27 at position 0 of the DnaK binding cleft (blue, left hand side) and bound with Ile 53 at position 0 (cyan, right hand side). The vertical blue line in panel F and the vertical cyan line in panel C are drawn at $^1$H chemical shifts of Ile 53 (where Ile 27 is at the 0 position) and Ile 27 (where Ile 53 is at the 0 position), respectively. These shifts are obtained from the fit. Solid curves are fits of the data to the four-state model depicted in *Figure 7*.

DOI: https://doi.org/10.7554/eLife.32764.011

The following figure supplements are available for figure 6:

**Figure supplement 1.** DrkN binding to DnaK/ATP but not to DnaK/ADP can be detected by $^1$H CEST.
DOI: https://doi.org/10.7554/eLife.32764.012

**Figure supplement 2.** DrkN SH3 binding to DnaK/ATP can be detected using magnetization exchange experiments.
DOI: https://doi.org/10.7554/eLife.32764.013

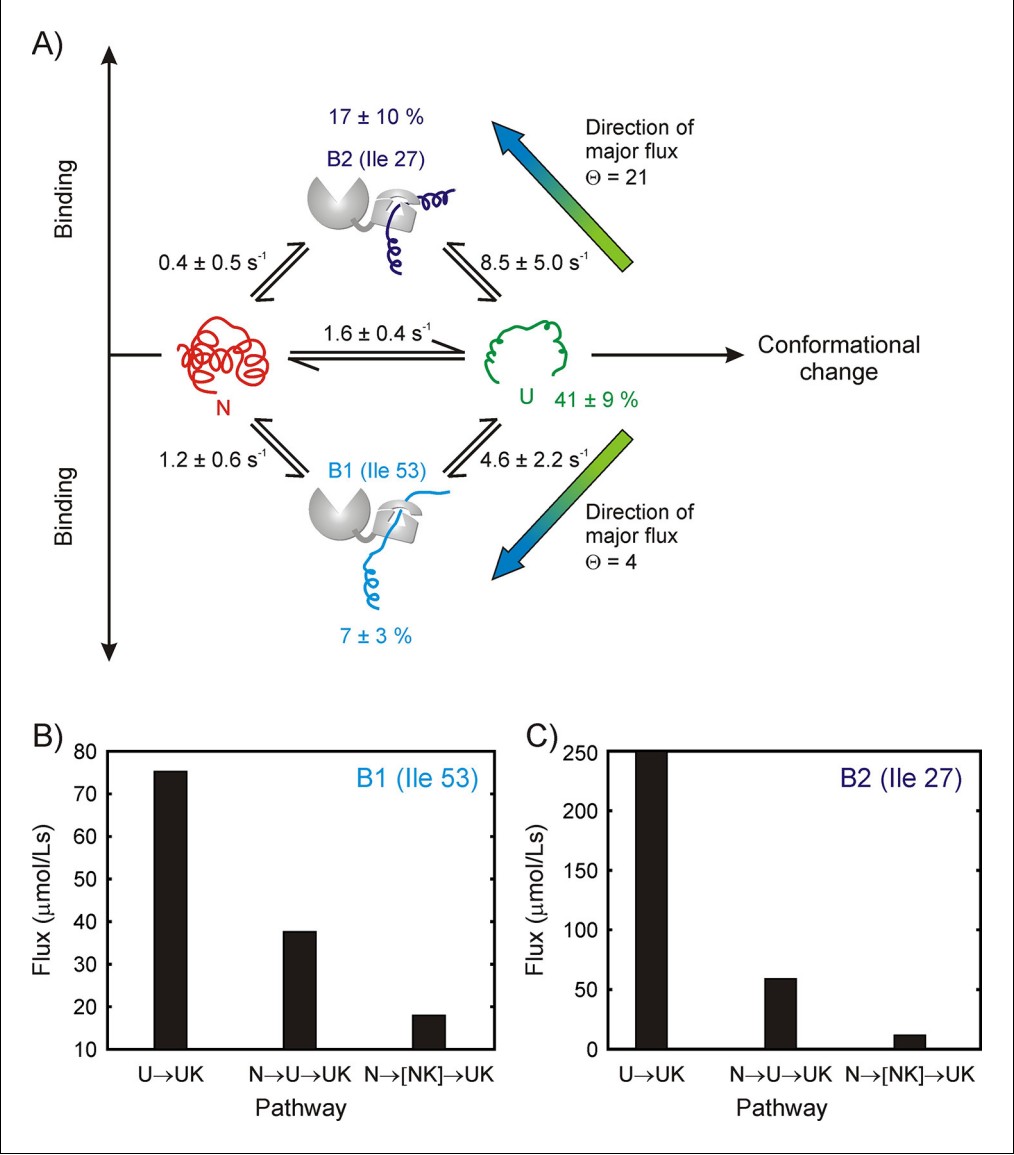

**Figure 7.** Binding model for the drkN SH3 – DnaK complex, along with parameter estimates and flux along different pathways. (**A**) Four-state model used for fitting $^1$H CEST data, comprising native (N), unfolded (U) and two DnaK-bound states, B1 and B2, corresponding to Ile 53 and Ile 27 at the central DnaK binding groove. Populations of the different states as well as rate constants for their interconversion are indicated. Mean values of parameters and errors are obtained as the averages and standard deviations from a bootstrap procedure. Values of $k_{ex}$ are shown ($k_{ex} = k_{UX} + k_{off}$), where $k_{UX}$ is an apparent pseudo-first order rate constant ($k_{UX} = k_{on}^{UX}[L]$) and [L] is the equilibrium concentration of free ligand (DnaK in this case). Directions of major fluxes for binding at sites 1 (Ile 53) and 2 (Ile 27) are indicated. Histograms showing flux values for the $U{\rightarrow}UK$, $N \rightarrow U \rightarrow UK$ and $N \rightarrow NK \rightarrow UK$ pathways leading to the formation of the two bound states B1 (**B**) and B2 (**C**).

DOI: https://doi.org/10.7554/eLife.32764.014

The following figure supplements are available for figure 7:

**Figure supplement 1.** Estimating the reliability of model parameters obtained by fitting $^1$H CEST data for SH3-DnaK binding to a four-state model.

DOI: https://doi.org/10.7554/eLife.32764.015

**Figure supplement 2.** Estimating parameter distributions using a bootstrapping procedure described in Materials and methods.

DOI: https://doi.org/10.7554/eLife.32764.016

B1 and B2 corresponding to Ile 53 and Ile 27 of drkN SH3 at position 0 of DnaK are found to be 4 and 21 respectively. *Figure 7B and C* show histograms of flux values for $U \to UK$, $N \to U \to UK$ and $N \to UK$ pathways. The flux through $N \to NK \to UK$ could not be estimated because $NK$ correlations are not observed, but an upper bound is given by the flow through $N \to UK$. Thus our data strongly support the notion that the dominant flux is through the $U \to UK$ pathway whereby DnaK directly selects the $U$ state, and therefore the binding mechanism can be described according to the CS model.

### Is the CS mechanism of interaction conserved across the Hsp70 family?

The Hsp70 family of chaperones are integral to the quality control machinery in organisms ranging from bacteria to humans and the Hsp70 client binding site sequence and architecture have been highly conserved through evolution. We thus wondered if the mechanism of client substrate recognition by Hsp70 is also conserved and whether a CS mode of binding might also apply to other Hsp70 chaperones. To this end we choose the constitutively expressed human Hsc70 for study and focused on the drkN SH3 substrate.

Upon addition of 500 µM U-$^2$H, Hsc70/ATP to a solution of 170 µM IM-$^{13}$CH$_3$ drkN SH3, no new peaks were observed (*Figure 8A,B*), suggesting that the affinity of drkN SH3 for Hsc70/ATP is lower than that for DnaK/ATP (*Figure 5*). A reduced affinity would lower the population of $UK$ and in the present case render it invisible to study by traditional NMR methods. For applications of this sort CEST is particularly powerful because the signals from sparsely populated protein conformations can be amplified by detecting them through NMR visible states (*Vallurupalli et al., 2017*). *Figure 8C,D* shows $^1$H CEST profiles derived from Ile 27 of $N$ and $U$ drkN SH3. Similar to our observations with DnaK, the native state CEST profile (red) shows a minor dip at the frequency of Ile 27 H$^{\delta 1}$ in the unfolded state, while the unfolded state profile (green) shows a pair of dips, with one at the position of the native state for Ile 27 H$^{\delta 1}$ and a second, initially unassigned. There are several lines of evidence that strongly indicate that the unassigned minor dip arises from Hsc70-bound drkN SH3. First the chemical shift of the dip, 0.35 ppm, is close to 0.2 ppm that was measured for the DnaK bound states B1 and B2 (*Figure 5*). Second, when CEST profiles are recorded of drkN SH3 bound to DnaK/ADP, where the exchange kinetics are much slower than for the ATP loaded chaperone form and too slow to observe exchange derived dips (see above), a minor dip in the region between 0.2–0.4 ppm is not observed in the unfolded state CEST profile (*Figure 6—figure supplement 1B*, middle). Finally, a $^{13}$C-$^1$H HMQC spectrum recorded after leaving the sample for approximately 2 days at room temperature, sufficient time for Hsc70/ATP to convert to Hsc70/ADP, shows a new peak at the $^1$H chemical shift of the minor dip, 0.35 ppm. This new peak belongs to Hsc70/ADP-bound drkN SH3 and is now visible because Hsc70/ADP has a higher affinity for substrate than Hsc70/ATP and the population of the bound state is higher. Notably, in the $N$ state CEST profile a small broad peak is observed at the position of Ile 27 H$^{\delta 1}$ in the bound state (~0.35 ppm). This dip is not the result of exchange between $N$ and $UK$ but can be attributed, instead, to an NOE between proximal residues Ile 27 and Ile 48 in the native drkN SH3 state and the fact that the bound state H$^{\delta 1}$ chemical shift for Ile 27 is degenerate with the $^1$H Ile δ1 shift of residue 48 in the native state. Confirmation that this is an NOE dip and not due to exchange is obtained by measuring CEST profiles of drkN SH3 bound to DnaK/ADP where an NOE dip is observed as well (*Figure 8E*) but, as discussed above, where the rates of binding/release are too slow for the development of minor dips resulting from exchange and by the observation of a distinct NOE crosspeak between Ile 27 and Ile 48 in a 3D $^{13}$C-$^{13}$C-$^1$H NOESY spectrum of ILVM-$^{13}$CH$_3$ drkN SH3. Taken together our results thus confirm that the major flux to $UK$ proceeds through $U$ so that human Hsc70 also employs a CS mode of interaction with the drkN SH3 client protein. As a final note, we have chosen not to quantify fluxes in this case because a bound state profile could not be obtained (peaks from the bound state are not observed in spectra) and because of the potential for contamination from NOE dips.

## Discussion

The Hsp70 chaperone plays a ubiquitous role in maintaining cellular proteostasis by recognizing and modulating the structure of its client proteins in an ATP-dependent manner to effect a variety of downstream functions that include protein folding, translocation and oligomer disassembly (*Balchin et al., 2016*). Structures of Hsp70 bound to a number of different peptide substrates have appeared (*Zhu et al., 1996*; *Zahn et al., 2013*; *Stevens et al., 2003*; *Clerico et al., 2015*) and both

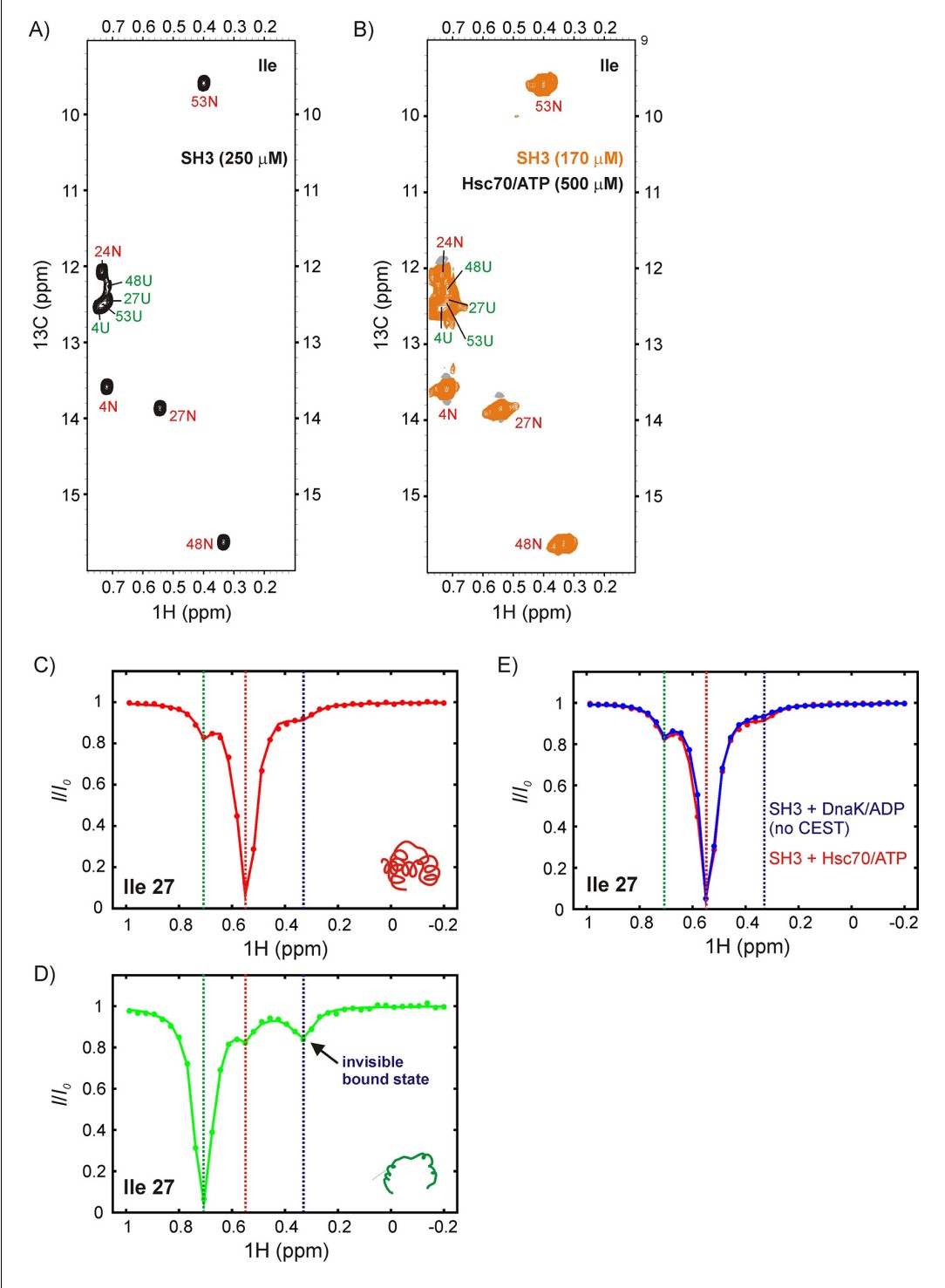

**Figure 8.** DrkN SH3 binds Hsc70 through the unfolded state. $^{13}C$-$^1H$ HMQC spectra of (**A**) 250 μM IM-$^{13}CH_3$ drkN SH3 without Hsc70 and (**B**) 170 μM IM-$^{13}CH_3$ drkN SH3 with 500 μM U-$^2H$ Hsc70/ATP. New peaks do not appear upon addition of Hsc70/ATP. Ile 27 $^1H^\delta$ CEST profiles from native (**C**) and unfolded (**D**) drkN SH3 in the presence of Hsc70/ATP acquired using the sample from panel B. The unfolded state CEST profile (panel D) shows a minor dip at the native state chemical shift (~0.55 ppm), but also a prominent second minor dip at ~0.33 ppm that arises from the binding of drkN SH3 to Hsc70/ATP. While the existence of the bound state cannot be discerned from the spectrum in panel B, binding is clearly revealed in the CEST profile. (**E**) The small dip observed in the native state CEST profile (panel C, drkN SH3 +Hsc70/ATP) at ~0.33 ppm is also present in the native state CEST profile of drkN SH3 +DnaK/ADP where binding/release to DnaK/ADP is too slow to generate exchange dips (see *Figure 6—figure supplement 1B*). This dip arises from cross-relaxation of Ile 27 with the nearby Ile 48 in the native state.
DOI: https://doi.org/10.7554/eLife.32764.017

biochemical and biophysical studies of Hsp70 with folding competent protein domains and intact proteins have been reported (*Kellner et al., 2014*; *Lee et al., 2015*; *Sekhar et al., 2015*; *Sekhar et al., 2016*; *Rodriguez et al., 2008*; *Sekhar et al., 2012a*; *Sekhar et al., 2012b*). However, the coupling between chaperone binding and substrate conformational changes remains poorly understood.

Understanding the mechanisms by which flexible molecules bind their targets has been the focus of considerable attention in the literature (*Boehr et al., 2009*; *Rauh et al., 2004*; *Zhang et al., 2007*; *Mittag et al., 2010*). In general, such molecules exist in an array of different interconverting conformations in solution at equilibrium, yet binding to a protein generally selects one of the states over the others. In principle, NMR spectroscopy is a valuable tool for obtaining mechanistic information about the binding process if the kinetics happen to be in a regime that is amenable to the methodology. In the case of zz-exchange this requirement puts exchange rates on the order of 1 s$^{-1}$ or larger, although in some cases significantly slower timescales have been studied using methionine methyl probes with very slow longitudinal relaxation rates (*Religa et al., 2010*). The CEST experiment, in general, is most sensitive to slightly faster processes with rates of at least 20–30 s$^{-1}$, but in the present application accurate estimates for rates of ~1 s$^{-1}$ could be obtained because profiles derived from spins in all of the interconverting states could be measured.

An elegant illustration of the utility of NMR in this regard was provided close to forty years ago in studies by Birdsall and coworkers using a one-dimensional analogue of the CEST approach described here (*Birdsall et al., 1980*). The authors were able to demonstrate that the enzyme dihydrofolate reductase (DHFR) selects one conformation of folinic acid, a product of the reaction with tetrahydrofolate that is catalyzed by DHFR, over a second, under conditions where both conformations exist in roughly equal proportions in solution in the absence of enzyme. Here we have extended this work by using state of the art NMR methods that include relaxation optimized techniques for recording spectra of methyl group probes (methyl TROSY) in high molecular weight complexes (*Rosenzweig and Kay, 2014*) and two-dimensional spin relaxation experiments such as CEST (*Vallurupalli et al., 2017*) and magnetization exchange (*Farrow et al., 1994*; *Palmer et al., 2001*) to probe the binding reaction of Hsp70 with cognate substrates. It is increasingly recognized that protein molecules, much like simpler small organic compounds such as folinic acid, exist in solution as an equilibrium mixture between multiple conformations, and that these diverse conformers may in some cases have different functional properties, including different propensities for binding to ligands. The targets of Hsp70 include an array of different protein substrates and elucidating what, if any, common structural features are recognized by the chaperone during the initial stages of binding to these targets would represent an important contribution towards understanding the overall binding mechanism.

We have studied the binding reaction of Hsp70 from *E. coli* (DnaK) to a pair of substrates with very different secondary structures, including the three-helical bundle domain R17* from chicken-brain α-spectrin and the all β-sheet SH3 domain from the Drosophila adaptor protein Drk. Both of these protein domains exist in solution as an equilibrium between $N$ and $U$ states, with peaks from each conformer observed in $^{13}$C-$^1$H correlation datasets so that probes are available that can report on preferred chaperone binding to either of these two predominant solution conformations. Relaxation based experiments establish that in both cases the dominant flux to the bound state, in which the substrate is unfolded in complex with DnaK (*Lee et al., 2015*), proceeds through the single step binding reaction $U \rightarrow UK$ as opposed to any scheme in which binding to $N$ is the first step in an $NK$ to $UK$ transition. In order to obtain insight into whether the DnaK binding mechanism is conserved among different Hsp70 orthologs we have carried out further studies using the constitutively expressed human Hsc70 protein and the drkN SH3 substrate. Notably, peaks from the bound state were not observed in spectra, reflecting the low affinity of the complex. However, an excited state dip was observed in the $^1$H CEST profile of Ile 27 H$^{\delta 1}$ in $U$ that could be assigned to the 'invisible' bound state, while a corresponding peak from chemical exchange was not observed for $N$. This establishes that the flux to the Hsc70 bound state, like that for DnaK, also proceeds via a $U \rightarrow UK$ mechanism. It is worth noting that all of the flux measurements described here were done at equilibrium, with the stability of samples monitored by recording a series of $^{13}$C-$^1$H correlation maps at different time points. It is, of course, possible to envision non-equilibrium measurements where the relative importance of different fluxes change over time as populations of states vary during the course of reactions, but this was not the case for the equilibrium studies conducted here.

Our data thus point to the conformational selection model as a good descriptor of Hsp70-ligand binding, at least for the substrates and chaperones that are considered in the present study. It is important to emphasize that the analysis of flux through IF and CS pathways discussed here is somewhat different than what is typically described in the literature where a comparison of fluxes through $N \rightarrow U \rightarrow UK$ (CS) and $N \rightarrow NK \rightarrow UK$ (IF) pathways exclusively is made, without considering binding proceeding directly from other equilibrium states ($U$ in this case) (**Hammes et al., 2009**; **Weikl and Paul, 2014**; **Vogt and Di Cera, 2012**). Here we have considered proteins as equilibrium ensembles, with binding reactions proceeding from any member of the ensemble, rather than focusing exclusively on the $N$ conformer as the initial state. An analysis that includes the possibility of fluxes from any of the existing equilibrium states of a protein in solution is required for distinguishing between different mechanisms of binding. In the case here $U$ is thermally accessible from $N$ and is significantly populated at equilibrium, with the $N \rightarrow U \rightarrow UK$ reaction contributing only a small amount to the net flux by which $UK$ is formed relative to the $U \rightarrow UK$ pathway for the substrates and Hsp70 chaperones considered.

The preference of Hsp70 for the unfolded state of the substrates demonstrates a 'holdase' recognition mechanism whereby Hsp70 captures intrinsic fluctuations in its client protein. Our results argue against a simple $N \rightarrow NK \rightarrow UK$ unfoldase scheme as a dominant binding mechanism, where the native state of the substrate is recognized by Hsp70 and induced to unfold upon binding. However, it must be emphasized that we have focused on elementary binding steps here and our results are not inconsistent with broader unfoldase mechanisms involving recognition of partially unfolded or misfolded regions of substrates that eventually result in unfolded Hsp70-bound clients via a multi-step process. For example, such a mechanism may be operative in the conversion of a misfolded state of luciferase to a globally unfolded, folding-competent ensemble (**Sharma et al., 2010**) where the elementary binding step is a holdase-like CS reaction, $U \rightarrow UK$, involving trapping by Hsp70 of local conformational fluctuations occurring in a small region of the protein.

A CS-based mechanism for Hsp70-substrate interactions is consistent with expectations based on Hsp70 function. First, such a binding mode ensures that Hsp70 will not randomly unfold properly folded and functional cellular proteins. Second, a number of cognate Hsp70 substrates such as nascent polypeptides synthesized on the ribosome (**Teter et al., 1999**) and proteins translocated across membranes (**Pilon and Schekman, 1999**) are unfolded and would therefore be poised for binding to Hsp70. Third, the CS binding mode ensures that Hsp70 will select only misfolded intermediates lacking stable secondary and tertiary structure for refolding rather than stably folded native conformations with exposed hydrophobic sidechains so that refolding occurs in a unidirectional manner from the misfolded to the native state. Finally, Hsp70 recognition sites on proteins such as $\sigma^{32}$ that bind in their native state are in long solvent-exposed loop regions (**Rodriguez et al., 2008**) where both sidechain and backbone motifs of the binding site residues are available for interaction.

It should also be noted that there are likely to be small differences in structure between the Hsp70-bound ($UK$) and the globally unfolded state ($U$) of substrates. Indeed, conformational sampling in the bound state is expected to be different from the globally unfolded state because each Hsp70 molecule divides the substrate into two distinct polypeptide segments at the binding site. We have shown earlier that tertiary interactions that are transiently present in the $U$ state of the unbound substrate are disrupted across the Hsp70 binding site (**Sekhar et al., 2016**), so that the two segments could have different overall structural propensities than would be the case for the full-length protein in its unfolded state. We cannot determine through our measurements whether in this case these small changes have occurred as a result of binding (IF) or prior to it (CS). Instead, our conclusions of CS being the prevalent mechanism, therefore, pertain to binding events involving interactions of the chaperone with states that have larger conformational differences such as $N$ and $U$.

The multiple Hsp70-bound conformations of R17* and drkN SH3 noted in this study reinforce our previous observations with the three-helix bundle substrate hTRF1 that Hsp70 binds promiscuously to its substrates, recognizing different sites containing aliphatic hydrophobic residues along the polypeptide sequence (**Rosenzweig et al., 2017**). Our results show that each binding site of the substrate can be recognized by Hsp70 when it becomes exposed to the solvent in the thermally accessible unfolded state, highlighting the prominent role played by conformational dynamics in this crucial protein-protein interaction.

# Materials and methods

**Key resources table**

| Reagent type (species) or resource | Designation | Source or reference | Identifiers | Additional information |
|---|---|---|---|---|
| Gene (*Gallus gallus*) | R17 | NA | Uniprot ID: P07751 | R17 is the 17th repeat of spectrin alpha chain |
| Gene (*Drosophila melanogaster*) | SH3 | NA | Uniprot ID: Q08012 | SH3 is the N-terminal SH3 domain of the Enhancer of Sevenless 2B |
| Gene (*Escherichia coli*) | DnaK | NA | Uniprot ID: P0A6Y8 | Hsp70 chaperone from E.coli |
| Gene (*Escherichia coli*) | GrpE | NA | Uniprot ID: P09372 | GrpE co-chaperone from E.coli |
| Gene (*Homo sapiens*) | Hsc70 | NA | Uniprot ID: P11142 | Heat shock cognate 71 kDa protein |
| Recombinant DNA reagent | pET29b(+) - R17* | This paper | | R17* is L90A R17, plasmid has tags etc. |
| Recombinant DNA reagent | pET28 - SH3 | This paper | | Plasmid has tags etc. |
| Recombinant DNA reagent | DnaK | DOI: 10.1073/pnas.1508504112 | | |
| Recombinant DNA reagent | GrpE | DOI: 10.1073/pnas.1508504112 | | |
| Recombinant DNA reagent | Hsc70 | DOI: 10.1073/pnas.1508504112 | | |
| Software, algorithm | NMRPipe | DOI: 10.1007/BF00197809 | | |
| Software, algorithm | Sparky | https://www.cgl.ucsf.edu/home/sparky/ | | |
| Software, algorithm | FuDA | http://www.biochem.ucl.ac.uk/hansen/fuda/ | | |
| Software, algorithm | ChemEx | https://github.com/gbouvignies/chemex | | |

## Plasmids, protein expression and purification

### R17*

The gene encoding L90A R17 (referred to here as R17*), containing an N-terminal hexa-His tag followed by a short linker and a Tobacco Etch Virus (TEV) protease cleavage site, was synthesized by Genscript and sub-cloned into a pET-29b(+) vector. U-$^2$H, Ileδ1-$^{13}$CH$_3$, Leu/Val-$^{13}$CH$_3$/$^{12}$CD$_3$, Met-$^{13}$CH$_3$ (ILVM-$^{13}$CH$_3$) labeling of R17* was carried out according to previously published protocols (*Tugarinov et al., 2006*; *Gelis et al., 2007*). Briefly, BL21(DE3) cells overexpressing R17* from the pET-29b(+) vector were grown at 37°C in D$_2$O minimal media containing $^{15}$NH$_4$Cl (1 g/L) and [$^2$H,$^{12}$C]-glucose (3 g/L) as the sole nitrogen and predominant carbon sources respectively. 60 mg/L 2-keto-3-d$_2$-4-$^{13}$C-butyrate, 80 mg/L 2-keto-3-methyl-d$_3$-3-d$_1$-4-$^{13}$C-butyrate and 100 mg/L methyl-$^{13}$CH$_3$-methionine precursors were added one hour before induction. Cells were induced at an OD$_{600}$ of 0.8 with 1 mM isopropyl β-D-1-thiogalactopyranoside (IPTG) and grown for 20 hr at 22°C. Cells were subsequently lysed and purified by affinity chromatography using a nickel-nitrilo triacetic acid (Ni-NTA) column. The His tag was cleaved with TEV protease at 4°C and R17* was separated from the tag and TEV protease by passing it over a Ni-NTA column. The protein was further purified on a Superdex 75 size exclusion column. R17* elutes as two peaks, both of which give identical $^1$H-$^{15}$N HSQC spectra. The protein content from both peaks was pooled for preparing NMR samples.

### M26L R17*

Ileδ1-$^{13}$CH$_3$, Met-$^{13}$CH$_3$-labeled M26L R17* was overexpressed and purified as described above for R17* with the following minor modifications. Cells were cultured in $^1$H$_2$O minimal media containing 3 g/L unlabeled glucose and 1 g/L $^{15}$NH$_4$Cl. 50 mg/L 2-keto-3-d$_2$-4-$^{13}$C-butyrate and 100 mg/L methyl-$^{13}$CH$_3$-methionine were used as precursors for labeling Ile (δ1) and Met methyl groups, respectively. Since the deuterons of the methylene group in the Ile precursor partially exchange to

$^1$H during protein overexpression, Ile residues of M26L R17* exist as mixtures of isotopomers and show characteristic 2- ($F_1$) and 3- ($F_2$) bond shifts in $^{13}$C-$^1$H correlation spectra. Chemical shifts of the isotopomers were not resolved in the NOESY spectrum that was recorded and thus were not problematic.

## DrkN SH3

The gene for the SH3 domain of *Drosophila melanogaster* Enhancer of sevenless 2B protein (drkN SH3) was cloned into the pET-28 vector using PCR amplification (Kapa Hifi, Kapa Biosystems, MA, U.S.A.) followed by Gibson assembly (New England Biolabs, MA, U.S.A.). The final gene incorporates a N-terminal Hexa-His tag followed by a TEV protease cleavage site. Selectively labeled, highly deuterated drkN SH3 was overexpressed in BL21(DE3) cells as described above using 60 mg/L 2-keto-3-d$_2$-4-$^{13}$C-butyrate and 100 mg/L methyl-$^{13}$CH$_3$-methionine without (IM-$^{13}$CH$_3$ labeled drkN SH3) or with 80 mg/L 2-keto-3-methyl-d$_3$-3-d$_1$-4-$^{13}$C-butyrate (ILVM-$^{13}$CH$_3$ labeled drkN SH3) as precursors. The protein was isolated from the lysate using a Ni-NTA column under denaturing (6 M guanidinium chloride) conditions. The unfolded protein was refolded on the column before elution by lowering the denaturant concentration stepwise from 6 to 4, 2, 1 and finally to 0 M. The His tag was removed by incubation with TEV protease overnight at 4°C. DrkN SH3 was separated from the tag and TEV protease by passing it over a Ni-NTA column and further purified on a Superdex 75 size exclusion column.

## Chaperones

Wild-type DnaK, T199A DnaK and T199A Hsc70 chaperones as well as the DnaK nucleotide exchange factor GrpE were overexpressed and purified as described previously (*Sekhar et al., 2015*). DnaK and Hsc70 were highly deuterated in all studies, with selective Ileδ1-$^{13}$CH$_3$, or Ileδ1-$^{13}$CH$_3$, Met-$^{13}$CH$_3$ labeling of DnaK carried out as described above. GrpE was overexpressed and purified from rich media as unlabeled protein.

## NMR samples

All NMR samples were prepared in 50 mM HEPES pH 8, containing 50 mM KCl, 5 mM MgCl$_2$, 1 mM EDTA and 1 mM TCEP in 100% D$_2$O with or without DnaK/ADP, DnaK/ATP or Hsc70/ATP. An ATP regeneration system composed of 50 units of creatine kinase, 20 mM phosphocreatine, was added to all samples containing DnaK/ATP or Hsc70/ATP. In the case of DnaK/ATP the regeneration system additionally contained 5 μM GrpE to facilitate nucleotide exchange (*Sekhar et al., 2015*).

## NMR data acquisition and analysis

Spectrometers ranging in $^1$H Larmor precession frequency (field strength) from 500 (11.7 T) to 800 MHz (18.8 T) were used for NMR data acquisition. All relaxation experiments (zz-exchange and CEST) were performed using a Bruker 800 MHz spectrometer equipped with a cryogenically cooled probe. The sample temperature was fixed across the different spectrometers by calibration using a thermocouple placed inside an NMR tube containing H$_2$O. All experiments were carried out at 25°C unless specified otherwise. NMR datasets were processed with the NMRPipe suite of programs (*Delaglio et al., 1995*) and visualized with Sparky (*Goddard and Kneller, 2006*).

## $^1$H CEST

$^1$H CEST datasets were acquired using a modified $^{13}$C-$^1$H HMQC pulse sequence incorporating a CEST exchange period of duration, $T_{EX}$, following the relaxation delay and immediately preceding the HMQC sequence (see *Figure 6—figure supplement 1A*). During $T_{EX}$ (150 ms) a weak saturating $^1$H B$_1$ field (20.3 Hz) was applied as a function of offset (one offset for each 2D dataset), spanning the range of Ile methyl $^1$H$^{δ1}$ chemical shifts (1.0 to −0.2 ppm). A 2 kHz $^{13}$C decoupling field (90$_x$-240$_y$-90$_x$, [*Freeman et al., 2011*; *Vallurupalli et al., 2012*]) was applied during $T_{EX}$ to eliminate one-bond $^{13}$C-$^1$H couplings that would otherwise appear in CEST profiles. 40 2D planes were acquired for each dataset, along with 32 scans and a relaxation delay of 1.5 s for a net recording time of 25 hr. Each $^1$H CEST dataset contained a reference plane that included B$_1$ irradiation far off-resonance for a duration $T_{EX}$.

In order to obtain robust fits of $^1$H CEST profiles estimates of $^1$H longitudinal relaxation rates are required. These were obtained by recording an experiment in which initial $^1$H magnetization was placed along the +z or -z axis in alternate scans, allowing relaxation to proceed for a period $T_{EX}$ during which a 2 kHz $^{13}$C decoupling field was applied, as for the CEST experiments. The resulting signal was subsequently read out using an HMQC pulse scheme. $T_{EX}$ was varied from 50 to 600 ms and peak intensities were extracted and fit to a single exponential decay function of the form $I = I_0 exp(-R_1 T_{EX})$ to obtain $^1$H $R_1$ values.

$^1$H chemical shift-dependent resonance intensities were extracted from the pseudo-3D dataset by fitting lineshapes globally across all 40 2D planes using the FuDA software package (http://www.bio-chem.ucl.ac.uk/hansen/fuda/). CEST profiles were subsequently fit to an appropriate model of chemical exchange using the program ChemEx (*Bouvignies, 2017*; copy archived at https://github.com/elifesciences-publications/chemex) to extract rate constants for the interconversion between states and their populations, that were then used in the calculation of flux values. Bootstrap analyses were performed (*Efron and Tibshirani, 1986*) to obtain mean values of rates, populations and errors. In each analysis 1000 bootstrapped datasets were generated, each of which contained CEST profiles from *N*, *U* and *B* states of Ile 27 and Ile 53. Data-points for each CEST profile in the bootstrapped dataset were selected randomly from each experimental CEST profile such that the total number of points per profile (*Daniels et al., 2014*) was retained. Each bootstrap dataset was then fit globally using ChemEx (*Bouvignies, 2017*) to obtain distributions of model parameters with the widths of the distributions taken as estimates of the errors in the parameter values.

## Magnetization exchange

Magnetization exchange experiments were performed using a $^{13}$C-$^1$H HMQC-based pulse sequence in which an exchange element of duration $T_{EX}$ was inserted immediately after the $^{13}$C evolution period. For R17* 13 2D planes were acquired with $T_{EX}$ ranging from 25 to 800 ms. $T_{MIX}$-dependent peak intensities for Met 26 were obtained by fitting lineshapes of *U*, *UK*, *U-UK* and *UK-U* peaks with FuDA (*Figure 4B*). The resulting profiles were globally fit to a two-state model of chemical exchange to extract exchange rate constants and populations. A bootstrap analysis similar to those described for $^1$H CEST data was carried out to evaluate errors in populations and rate constants.

## $^{13}$C-$^{13}$C-$^1$H NOESY

3D $^{13}$C-edited NOESY datasets recording correlations of the type ($^{13}C_j$-NOE-$^{13}C_k$-$^1H_k$) were acquired on samples of R17* (400 µM) or drkN SH3 (250 µM) bound to DnaK/ADP (800 µM for R17* and 500 µM for drkN SH3) using a modified HMQC-based version of a previously published pulse sequence (*Zwahlen et al., 1998*). Mixing times of 200 ms were used in both cases.

## Acknowledgements

This work was supported by grants from the Natural Sciences and Engineering Research Council of Canada and the Canadian Institutes of Health Research (LEK), as well as funding from the Azrieli Foundation, the Blythe Brenden-Mann New Scientist Fund, and the Roshan Family Foundation (RR). LEK holds a Canada Research Chair in Biochemistry. We dedicate this paper to the memory of the late Alex D Bain who served both as a mentor as we attempted to understand the nuances of chemical exchange and as a good friend whose encouragement and good cheer was always greatly appreciated.

## Additional information

### Competing interests

Lewis E Kay: LEK is a Reviewing Editor in eLife. The other authors declare that no competing interests exist.

## Funding

| Funder | Author |
| --- | --- |
| Canadian Institutes of Health Research | Lewis E Kay |
| Natural Sciences and Engineering Research Council of Canada | Lewis E Kay |
| Azrieli Foundation | Rina Rosenzweig |
| Blythe Brenden-Mann New Scientist Fund | Rina Rosenzweig |
| Roshan Family Foundation | Rina Rosenzweig |

The funders had no role in study design, data collection and interpretation, or the decision to submit the work for publication.

## Author contributions

Ashok Sekhar, Conceptualization, Resources, Data curation, Software, Formal analysis, Supervision, Validation, Investigation, Visualization, Methodology, Writing—original draft, Project administration, Writing—review and editing; Algirdas Velyvis, Guy Zoltsman, Resources, Investigation; Rina Rosenzweig, Conceptualization, Resources, Supervision, Investigation; Guillaume Bouvignies, Resources, Data curation, Software, Formal analysis, Writing—review and editing; Lewis E Kay, Conceptualization, Data curation, Formal analysis, Supervision, Funding acquisition, Validation, Investigation, Methodology, Writing—original draft, Project administration, Writing—review and editing

## Author ORCIDs

Ashok Sekhar http://orcid.org/0000-0002-8628-7799
Guillaume Bouvignies http://orcid.org/0000-0003-4398-0320
Lewis E Kay http://orcid.org/0000-0002-4054-4083

## Decision letter and Author response

Decision letter https://doi.org/10.7554/eLife.32764.030
Author response https://doi.org/10.7554/eLife.32764.031

# Additional files

## Supplementary files

• Transparent reporting form
DOI: https://doi.org/10.7554/eLife.32764.018

## Major datasets

The following previously published datasets were used:

| Author(s) | Year | Dataset title | Dataset URL | Database, license, and accessibility information |
| --- | --- | --- | --- | --- |
| Zuiderweg ERP, Bertelsen EB | 2009 | NMR-RDC / XRAY structure of E. coli HSP70 (DNAK) chaperone (1-605) complexed with ADP and substrate | https://www.rcsb.org/pdb/explore/explore.do?structureId=2kho | Publicly available at the RCSB Protein Data Bank (Accession no: 2KHO) |
| Zhu X, Zhao X, Burkholder WF, Gragerov A, Ogata CM, Gottesman ME, Hendrickson WA | 1996 | The substrate binding domain of dnak in complex with a substrate peptide, determined from type 1 native crystals | https://www.rcsb.org/pdb/explore/explore.do?structureId=1dkz | Publicly available at the RCSB Protein Data Bank (Accession no: 1DKZ) |
| Kopp J, Mayer MP, Sinning I | 2012 | Open conformation of ATP-bound Hsp70 homolog DnaK | https://www.rcsb.org/pdb/explore/explore.do?structureId=4b9q | Publicly available at the RCSB Protein Data Bank (Accession no: 4B9Q) |

| Grum VL, Li D, MacDonald RI, Mondragon A | 1999 | Crystal structure of repeats 16 and 17 of chicken brain alpha spectrin | https://www.rcsb.org/pdb/explore/explore.do?structureId=1cun | Publicly available at the RCSB Protein Data Bank (Accession no: 1CUN) |
| Forman-Kay JD, Bezsonova I, Singer A, Choy W-Y, Tollinger M | 2005 | Solution structure of the N-terminal SH3 domain of DRK | https://www.rcsb.org/pdb/explore/explore.do?structureId=2a36 | Publicly available at the RCSB Protein Data Bank (Accession no: 2A36) |

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
