## [Decision Letter]

Thank you for submitting your article "Conserved conformational selection mechanism of Hsp70 chaperone-substrate interactions" for consideration by *eLife*. Your article has been reviewed by three peer reviewers, one of whom, Volker Dötsch (Reviewer #1), is a member of our Board of Reviewing Editors, and the evaluation has been overseen by John Kuriyan as the Senior Editor. The following individuals involved in review of your submission have agreed to reveal their identity: Hashim Al-Hashimi (Reviewer #2); Charalampos Babis Kalodimos (Reviewer #3).

The reviewers have discussed the reviews with one another and were enthusiastic about the presented data. The Reviewing Editor has drafted this decision to help you prepare a revised submission.

Summary:

Hsp70 plays a central role in protein homeostasis. The Hsp70 chaperone machinery has been extensively studied over the years. However, due to technical challenges some key questions remain elusive. One fundamental question is how Hsp70 binds to an unfolded substrate. Does Hsp70 bind to already exposed unfolded segments (conformational selection) or actively change the conformation (induced fit) of the substrate using the energy from ATP hydrolysis. Sekhar et al. used cleverly designed NMR approaches to address this question. Using ZZ-exchange and CEST experiments, the authors were able to discriminate between the conformational selection and induced fit mechanisms. By studying the interaction with two model substrates and both the *E. coli* and human Hsp70 the authors found a conserved conformational selection mechanism of Hsp70-substrate interactions. The approach presented here can in principle be used to characterize other chaperone systems or protein machines.

Essential revisions:

1) In building the CS and IF models, the authors indicate that the *U* and *N* states are themselves ensembles. Are there any assumptions being made regarding the various microscopic affinities and rates of interconversion for individual conformers within the ensembles? For example, is it assumed that the interconversion among members of the ensemble is much faster than ligand binding or *U* to *N* transitions?

2) Related to this question:

Are the affinities and *N* to *U* rates averages over individual conformers in the ensembles? What features does a given conformation have to have to be included as a separate species in the reaction mechanism?

3) Along similar lines, the authors clearly point out some of the assumptions pertaining to the similarity between the *U* and *UK* states. Under what conditions do the structures diverge sufficiently that the presented model breaks down and one has to implicitly include a third species e.g. *U** in the reaction?

4) Regarding the statement "Our analysis of flux through IF and CS pathways here is thus somewhat different than what is typically probed in the literature where a comparison of fluxes through *N* to *U* to *UK* (CS) and *N* to *NK* to *UK* (IF) pathways exclusively is made, without considering binding proceeding directly from other equilibrium states (*U* in this case)" I think this is the case because of the unusually high population of the alternative conformation in the free state as well as low rate of kinetic interconversion between *N* and *U*? Perhaps the authors could clarify this a little more.

---

## [Author Response]

Essential revisions:1) In building the CS and IF models, the authors indicate that the U and N states are themselves ensembles. Are there any assumptions being made regarding the various microscopic affinities and rates of interconversion for individual conformers within the ensembles? For example, is it assumed that the interconversion among members of the ensemble is much faster than ligand binding or U to N transitions?

Yes, we have assumed that the interconversion between *N* and *U* sub-ensembles is faster than *N-U* exchange or ligand binding.

2) Related to this question:Are the affinities and N to U rates averages over individual conformers in the ensembles? What features does a given conformation have to have to be included as a separate species in the reaction mechanism?

The affinities and rates that are reported are indeed averages over conformations in a sub-ensemble of *N* or *U*. In order for a given conformation to be included as a separate species in the model, it has to be visible either in the NMR spectrum with distinct chemical shifts, or observable in the CEST profile as a minor dip.

3) Along similar lines, the authors clearly point out some of the assumptions pertaining to the similarity between the U and UK states. Under what conditions do the structures diverge sufficiently that the presented model breaks down and one has to implicitly include a third species e.g. U* in the reaction?

Additional species such as alternative unfolded states *U** would be included in the model if they have distinct chemical shifts (if they are significantly populated), or are reflected in CEST profiles as minor dips (if they are sparsely populated). As a corollary in our particular case, we do not include *NK* as a separate species because we see no evidence in NMR data for its formation. This is articulated in the revised manuscript where we write:

“The intermediate state along the IF pathway, corresponding to *NK*, is not included in the model because there is no evidence from ^13^C-^1^H HMQC spectra of SH3/DnaK samples or from the CEST profiles justifying the explicit inclusion of such a state; instead, the three-state *N-NK-UK* pathway was approximated as *N-UK*.”

In response to the related queries 1-3, we have modified the manuscript to include a paragraph that outlines what the assumptions are. In the revised manuscript we now write:

"The *N* and *U* states defined above may themselves be composed of sub-ensembles of interconverting conformers, for example {*N_1_,N_2_*,…} and {*U_1_,U_2_*,…}. In constructing the CS and IF models in terms of *N* and *U* states it is assumed that the interconversion between sub-states *N_i_* or between *U_i_* occurs much faster than the *N-U* exchange or than ligand binding, with a rate that is fast on the NMR chemical shift timescale. In this manner only single distinct NMR peaks are observed for spins in *N* or *U* states rather than separate peaks for each element of the sub-ensembles (*N_i_*) or (*U_i_*) so that *N* or *U* can thus be treated as single ‘averaged’ conformers. It naturally follows that the affinities and rates obtained from NMR experiments will be averages over the members of each sub-ensemble. It is also the case that additional intermediates such as alternate unfolded states (*U**) can be explicitly included in any model of exchange only if their presence is detected either by distinct peaks in NMR spectra or by minor dips in CEST profiles."

4) Regarding the statement "Our analysis of flux through IF and CS pathways here is thus somewhat different than what is typically probed in the literature where a comparison of fluxes through N to U to UK (CS) and N to NK to UK (IF) pathways exclusively is made, without considering binding proceeding directly from other equilibrium states (U in this case)" I think this is the case because of the unusually high population of the alternative conformation in the free state as well as low rate of kinetic interconversion between N and U? Perhaps the authors could clarify this a little more.

In our case, as the reviewer states, *U* is thermally accessible and significantly populated at equilibrium, so that most of the flux from the free to DnaK-bound R17* flows through the U-UK pathway. Accordingly, there is no reason to ignore this flux contribution. A similar scenario also occurs in the context of binding of the SH3 domain to DnaK. We discuss this in some detail in the revised text:

“Our data thus point to the conformational selection model as a good descriptor of Hsp70-ligand binding, at least for the substrates and chaperones that are considered in the present study. It is important to emphasize that the analysis of flux through IF and CS pathways discussed here is somewhat different than what is typically described in the literature where a comparison of fluxes through N→U→UK (CS) and N→NK→UK(IF) pathways exclusively is made, without considering binding proceeding directly from other equilibrium states (*U* in this case) (Hammes, Chang and Oas, 2009; Weikl and Paul, 2014; Vogt and Di Cera, 2012). Here we have considered proteins as equilibrium ensembles, with binding reactions proceeding from any member of the ensemble, rather than focusing exclusively on the *N* conformer as the initial state. An analysis that includes the possibility of fluxes from any of the existing equilibrium states of a protein in solution is required for distinguishing between different mechanisms of binding. In the case here *U* is thermally accessible from *N* and is significantly populated at equilibrium, with the N→U→UK reaction contributing only a small amount to the net flux by which *UK* is formed relative to the U→UK pathway for the substrates and Hsp70 chaperones considered.”

It is important to emphasize, however, that even if the *U* state was only sparely populated relative to *N* this does not mean that one should ignore it as a potential starting point for binding. Indeed, the flux from *A* to *B* depends not only on the concentration of *A* but also on the rate constant from *A* to *B, k_AB_*. If *k_AB_* >> *k_CB_* then the flux from *A* to *B* can be much larger than from *C* to *B* even if [C] > [A]. In principle all equilibrium states should be considered in establishing the major pathway for a given reaction.

The reviewers also ask about the low rate of exchange between *U* and *N* facilitating a separate comparison of fluxes through these starting points. Indeed they are correct. We illustrate this in Figure 2 where we show in Figure 2 that in the limit of fast interconversion between *U* and *N* (relative to the kinetics of binding) no distinction in flux pathways can be made, while when the exchange is slow relative to binding (Figure 2) binding from *U* or *N* can indeed be distinguished.